# Bridging Sign and Spoken Languages: Pseudo Gloss Generation for Sign Language Translation

**Jianyuan Guo**[1][*]    **Peike Li**[2]    **Trevor Cohn**[2]

[1] City University of Hong Kong    [2] Google

## Abstract

Sign Language Translation (SLT) aims to map sign language videos to spoken language text. A common approach relies on gloss annotations as an intermediate representation, decomposing SLT into two sub-tasks: video-to-gloss recognition and gloss-to-text translation. While effective, this paradigm depends on expert-annotated gloss labels, which are costly and rarely available in existing datasets, limiting its scalability. To address this challenge, we propose a gloss-free pseudo gloss generation framework that eliminates the need for human-annotated glosses while preserving the structured intermediate representation. Specifically, we prompt a Large Language Model (LLM) with a few example text-gloss pairs using in-context learning to produce draft sign glosses from spoken language text. To enhance the correspondence between LLM-generated pseudo glosses and the sign sequences in video, we correct the ordering in the pseudo glosses for better alignment via a weakly supervised learning process. This reordering facilitates the incorporation of auxiliary alignment objectives, and allows for the use of efficient supervision via a Connectionist Temporal Classification (CTC) loss. We train our SLT model—consisting of a vision encoder and a translator—through a three-stage pipeline, which progressively narrows the modality gap between sign language and spoken language. Despite its simplicity, our approach outperforms previous state-of-the-art gloss-free frameworks on two SLT benchmarks and achieves competitive results compared to gloss-based methods.

## 1 Introduction

Sign languages are the primary means of communication for the deaf and hard-of-hearing communities [47], functioning as independent linguistic systems with unique lexicons, grammars and syntax [4]. Unlike spoken languages, they convey meaning using a combination of manual gestures such as hand shapes, movements, and positioning, and non-manual cues, including facial expression, mouthing and eyebrow movements [33]. Despite their crucial role in communication, sign languages often lack standardized written forms (so-called *glossing*, which we discuss shortly), creating significant challenges for both accessibility and automatic translation. Sign Language Translation (SLT) has emerged as a key research area aimed at converting sign language videos into textual representation [32, 9, 8, 57, 10]. However, for many reasons, ranging from the complexity of the input (video) to their linguistic complexity, SLT remains a challenging problem within the field of computer vision and natural language processing.

The task of SLT is typically framed as a sequence-to-sequence problem, taking video input of a signer, which is processed by vision encoder, and then generating a text output in written language (e.g., English) using a translation decoder, such as an LSTM or Transformer. These approaches can be broadly categorized into gloss-based and gloss-free methods, based on whether they incorporate gloss

---

[*]Work done during an internship at Google. Correspondence to `jianyguo@cityu.edu.hk`

| Dataset | Gloss | Language | Duration (h) | | | | Vocabulary (K) | | | | SOTA BLEU4 | | |
|---|---|---|---|---|---|---|---|---|---|---|---|---|---|
| | | | train | val | test | total | train | val | test | total | Gloss-based | Gloss-free | Δ |
| Phoenix14T [8] | ✓ | DGS | 9.2 | 0.6 | 0.7 | 11 | 2 | 0.9 | 1 | 2.9 | 29.0[†] [20] | 24.3[†] [26] | 4.7 |
| CSL Daily [60] | ✓ | CSL | 20.6 | 1.2 | 1.4 | 23 | 2 | 1.3 | 1.3 | 2.3 | 25.8[†] [11] | 20.6[†] [26] | 5.2 |
| BOBSL [3] | ✓ | BSL | 1236 | 20 | 204 | 1467 | 72 | 14 | 35 | 78 | 7.3[‡] [27] | 3.3 [27] | 4.0 |
| How2Sign [16] | ✗ | ASL | 69.6 | 3.9 | 5.6 | 79 | 15.6 | 3.2 | 3.6 | 16 | - | 15.5 [45] | - |
| OpenASL [46] | ✗ | ASL | - | - | - | 288 | - | - | - | 33 | - | 21.2 [28] | - |
| Youtube-ASL [51] | ✗ | ASL | - | - | - | 984 | - | - | - | 60 | - | - | - |

Table 1: Comparison of sign language translation datasets. [†] Results are sourced from Papers With Code as of April 2025 (large-scale pre-training methods [38] are excluded). [‡] Indicates the *oracle setting* where previous ground truth sentences are used in [27].

annotations. Glosses are a written representation of sign language, denoting the sequences of signs used, which often differs substantially from the words used in the written language text, in terms of word choice, grammar and ordering, alongside sign-language specific concepts such as pronominal referents, classifiers, and non-manual signs. Overall, as a linguistic description of sign language, glosses should provide a much richer source of supervision for understanding signed video input compared to the use of written language (closed captions).

Gloss-based methods typically involve pre-training the visual encoder on Sign Language Recognition (SLR) [13, 43], converting videos into gloss sequences. The task then becomes a standard machine translation problem, mapping these glosses to spoken language text. However, these methods are limited by their reliance on expert-annotated gloss labels, which are time-consuming and costly to produce, as glossing is not widely used in the deaf community. Accordingly, glosses are scarce in existing datasets [16, 51, 46]. In contrast, gloss-free methods [10, 58] have gained attention for directly training models on sign videos and corresponding spoken language text, bypassing the need for gloss annotations. While these methods streamline the pipeline and reduce annotation burdens, they face a performance gap compared to gloss-based approaches, as shown in Table 1. This discrepancy arises from the inherent differences between sign and spoken languages, making direct translation more challenging. In this context, gloss annotations remain vital for bridging these linguistic gaps and ensuring effective translation.

Recent approaches seek to leverage approximate *pseudo glosses* to bridge the semantic gap between visual and textual features [59, 28]. For example, Sign2GPT [53] synthesizes pseudo glosses by filtering out function words based on their part-of-speech tags, retaining only nouns, adjectives, and numerals. However, real gloss sequences often contain terms not in the original text, and their order often differs from the written text. This misalignment between pseudo-glosses and video inputs complicates the use of conventional training methods, such as Connectionist Temporal Classification (CTC) loss [19], reducing their effectiveness.

To combine the advantages of gloss-free methods with improved performance, we propose a framework that generates pseudo glosses using LLMs. This approach enables us to decompose the SLT process into pre-training and fine-tuning stages, as illustrated in Figure 1, bridging the gap between visual signals and spoken language, and allowing the SLT model to leverage gloss-like structures for enhanced performance. The core idea is that the semantic information of sign language is inherently embedded in reference translations provided alongside the video in standard SLT datasets, making it possible to systematically derive approximate glosses from it. By harnessing the reasoning capabilities of LLMs [49, 1], we extract key information from the text to generate pseudo glosses. To further improve this process, we utilize the in-context learning ability of LLMs, providing a small set of example text-gloss pairs (*e.g.*, 30). This allows the model to better capture the underlying structure and generate pseudo-glosses that align with sign representations.

To mitigate the potential misalignment between LLM-generated glosses and the actual sign sequences in videos, we introduce a reordering method, framed as a weakly supervised learning problem. We train an order-invariant classifier to predict the set of glosses in each video, which is then used to infer a temporal alignment. Armed with reordered pseudo glosses, we decompose the SLT task into three key stages [10]: video to pseudo gloss recognition (Sign2Gloss), pseudo gloss to text translation (Gloss2Sign), and final fine-tuning (Sign2Text), as illustrated in Figure 1. This structured breakdown alleviates the challenges of direct video-to-text mapping and improves translation performance. This structured breakdown follows a curriculum learning [5] paradigm, where the model gradually progresses from simpler synthetic tasks to more challenging real-world translation tasks, thereby alleviating the challenges of direct video-to-text mapping and enhancing overall translation

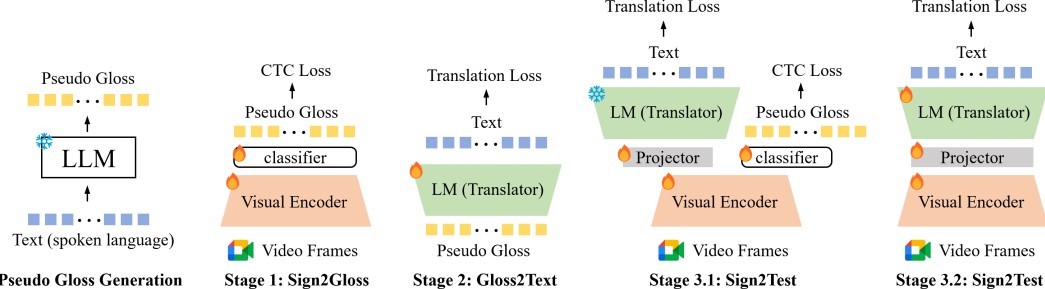

Figure 1: The training pipeline comprises three stages. Pseudo glosses are generated by LLMs and reordered using weakly supervised learning paradigms (see Sec. 3).

performance. Experiments demonstrate that our method outperforms existing gloss-free approaches and provides a viable alternative to traditional gloss-based methods.

## 2   Related Work

**Gloss Supervision in SLT.** Gloss-based SLT employs an intermediate textual representation, known as gloss, to bridge sign video sequences and spoken language text. Gloss provide a precise, temporally aligned transcription of sign video sequences, offering explicit linguistic supervision that facilitates the learning of structured representations in SLT models. Early works [60, 8] adopted a two-stage approach, where SLT is decomposed into gloss recognition [15, 35] followed by machine translation [30, 2, 41]. Inference proceeds in two steps: a *Sign2Gloss* module first predicts the gloss sequence, which is then translated into text by a *Gloss2Text* module. Later studies [10, 11] revealed that an end-to-end *Sign2Text* model can outperform the traditional *Sign2Gloss2Text* pipeline by jointly optimizing both stages. However, the intermediate gloss remains essential for pre-training *Sign2Gloss* and *Gloss2Text* modules, aiding in disentangling visual feature extraction from language modeling and enhancing structured learning.

**Gloss-free SLT Framework.** While gloss annotations offer precise alignment of keywords with sign videos, they are costly and require expert annotators with domain knowledge to transcribe videos at the lexical level. Consequently, many large-scale sign language datasets [16, 46, 51] lack gloss annotations. Recent methods [28, 56, 37, 59, 53] have explored gloss-free SLT by learning implicit vision-text alignment without explicit gloss supervision. One line of research leverages pseudo-labeling techniques to approximate gloss supervision from raw video-text pairs. For example, GFSLT-VLP [59] pre-trained the visual encoder and language decoder using a CLIP-based contrastive learning method. Sign2GPT [53] heuristically retained only content words (noun, numeral, adjective, *etc.*) from the text to generate glosses, which are then used to pre-train the visual encoder. VAP [28] utilized CLIP-based word representations, pre-trained on large-scale corpora, to improve the alignment between visual frames and lexical words in the paired text. Other studies aimed to scale up the amount of training data [51, 38] and employed stronger pre-training strategies [45] to enhance SLT models.

Unlike previous gloss generation methods employing contrastive learning to roughly align frames with corresponding words in text, we propose to use LLMs to generate pseudo glosses[2]. Sign language possesses its own grammar, vocabulary, and expression patterns. Leveraging in-context learning and linguistic understanding of spoken language (text), LLMs can akin to a human expert, infer reliable gloss-like annotations from a text when provided with some gloss-text examples. These generated glosses serve as effective supervision signals, enhancing alignment between visual encoder and language decoder during the training of SLT model.

**Pre-training for SLT.** Another research direction focuses on enhancing SLT performance through pre-training. SignBERT [25] first proposed a self-supervised pre-training followed by downstream-task fine-tuning framework. Building on this, SignBERT+ [24] modeled hand poses as visual tokens using multi-level masked modeling strategies and incorporated model-aware hand priors to capture hierarchical context. Meanwhile, SHuBERT [21] adapted masked token prediction objectives to multi-

---

[2]Our generated glosses are not video-derived *true* glosses (should only be translated from videos). Instead, we leverage LLMs' strength in key info extraction, as core content is reflected in the corresponding spoken text.

stream visual sign language inputs, learning to predict multiple targets corresponding to clustered hand, face, and body pose streams. SSVP-SLT [45] leveraged MAE [23, 22] to pre-train a capable sign language encoder on video data alone without gloss or text annotations, achieving stat-of-the-art results on How2Sign.

**Weakly Supervised Sequence Alignment.** Connectionist Temporal Classification (CTC) [19] loss is widely used in speech recognition and sequence modeling, allowing training without frame-level alignments. In our case, LLM-generated pseudo glosses may not match the actual temporal order of the signing sequence. To address this, we treat these pseudo glosses as an unordered label set and align them with the video using weakly supervised learning [6, 17, 34]. This process learns the correct temporal alignment before applying the CTC loss. Similar weakly supervised methods have been explored in action segmentation [7, 36, 44], where only unordered action labels are available, with no temporal boundaries or ordering information. This line of work is closely related to our objective, where we seek to recover the temporal structure of LLM-generated glosses from unordered supervision to enhance alignment between the visual encoder and language decoder.

# 3 Methodology

In this section, we first revisit previous gloss-based methods that decompose SLT into three training stages in Sec. 3.1. Then, we introduce our pseudo gloss generation method in Sec. 3.2. We describe how to reorder the pseudo glosses via weakly supervised learning in Sec. 3.3. Finally, we present the overall training and inference details of our approach in Sec. 3.4.

## 3.1 Preliminaries: Gloss-based Frameworks

Given an input sign video $\mathcal{V} = \{v_1, ..., v_T\}$ with $T$ frames, the objective is to learn a neural network $N_\psi(\cdot)$ that directly translates the sign video into the corresponding spoken language sentence $\mathcal{S} = \{s_1, ..., s_U\}$ with $U$ words (punctuation marks are omitted here):

$$\mathcal{S} = N_\psi(\mathcal{V}), \tag{1}$$

To facilitate learning, prior works [10, 11] have decomposed SLT into three sub-tasks: (1) a visual recognition task that converts sign videos into glosses (*Sign2Gloss*), (2) a translation task that maps glosses to spoken language texts (*Gloss2Text*), and (3) an end-to-end SLT model that integrates both steps (*Sign2Text*). This hierarchical structure enables pre-training of the first two tasks independently before fine-tuning the joint SLT model.

**Stage 1: *Sign2Gloss*.** The goal of this stage, also known as sign language recognition (SLR), is to predict the gloss sequence $\mathcal{G} = \{g_1, ..., g_M\}$ with $M$ glosses from the sign video $\mathcal{V}$. This is achieved using a vision encoder $\psi_V$, trained with the following objective:

$$\mathcal{L}_{SLR} = -\log p(\mathcal{G}|\mathcal{V}; \psi_V), \tag{2}$$

**Stage 2: *Gloss2Text*.** Given the gloss sequence $\mathcal{G}$, a natural language translation model $\psi_T$ generates the spoken language sentence $\mathcal{S}$, trained with the objective:

$$\mathcal{L}_{SLT} = -\log p(\mathcal{S}|\mathcal{G}; \psi_T). \tag{3}$$

**Stage 3: *Sign2Text*.** To enable end-to-end fine-tuning, an additional randomly initialized projector $\phi$ is used to map the visual features into the latent space of the translation model's input. This allows for joint optimization of both recognition and translation, with the overall objective formulated as:

$$\mathcal{L} = \mathcal{L}_{SLT} + \mathcal{L}_{SLR} = -\log p(\mathcal{S}|\mathcal{V}; \psi_T, \psi_V, \phi) - \log p(\mathcal{G}|\mathcal{V}; \psi_V). \tag{4}$$

Usually, the CTC loss [19] is employed to compute $p(\mathcal{G}|\mathcal{V})$, while a cross-entropy (CE) loss with next-word prediction is used for $p(\mathcal{S}|\mathcal{G})$ and $p(\mathcal{S}|\mathcal{V})$. More details can be found in the appendix.

Traditional SLT methods heavily rely on gloss annotations to align visual and textual representations. However, in a gloss-free setting, $\mathcal{G}$ is unavailable, posing a significant challenge to this paradigm. To address this, we propose PGG-SLT, an annotation-efficient (applicable to most gloss-free settings)[3] framework that does not require gloss annotations.

---

[3]Specifically, our approach is not strictly gloss-free on PHOENIX14T dataset, as we use a small portion of available true gloss to generate pseudo-glosses. For datasets like How2Sign that inherently lack gloss annotations, we use gloss-text examples from other datasets, making our method gloss-free relative to the target dataset.

## 3.2 Generating Pseudo Gloss via Large Language Model

The glosses provide more concise and direct representations of information that align with the video, compared to the full text. More importantly, their content remains semantically grounded in the original text, enabling a systematic extraction of key information to generate pseudo glosses. As shown in Table 2, certain function words in the text (*e.g.*, articles and auxiliary verbs) are often redundant in glosses and can be omitted without affecting the conveyed meaning.

Fortunately, well-trained multilingual large language models (LLMs) can serve as surrogate experts proficient in various sign languages, and are capable of performing the basic tasks of glossing such as filtering out unimportant words while preserving key content. Furthermore, by providing a few reference text-gloss example pairs, we can harness the in-context learning ability of LLMs to generalize the gloss extraction process. Even with a minimal number of examples (*e.g.*, just 30 pairs from Phoenix14T, accounting for no more than 0.4% of the training split), the LLM can accurately identify and structure essential content words that convey meaningful information. Additionally, it can perform lemmatization, ensuring verb normalization and consistency across different tenses. This approach enables the efficient generation of reasonable quality pseudo glosses (refer to as **LLM Gloss**) without requiring explicit linguistic annotations or expert intervention. More details about the prompts are illustrated in Figure 3, Figure 4, and Figure 6 in appendix.

| Phoenix14T [8] | Text | im Nordwesten und Westen gibt es vorübergehend auch gefrierenden Regen. (in the northwest and west there is temporarily also freezing rain.) |
|---|---|---|
| | Gloss | REGION (region) / AUCH (also) / FROST (frost) / REGEN (rain) / MIT (with) / FROST (frost) |
| Belebele [14] | Text | Fish often die because of the high concentrations of the toxin in the waters. |
| | Gloss | FISH / OFTEN / DIE / WHY / TOO / MUCH / T-O-X-I-N / IN / WATER |

Table 2: Example text and gloss from German Sign Language (DGS) [8] and American Sign Language (ASL) [14]. The dashes in 'toxin' denote finger spelling.

## 3.3 Reorder of LLM-Generated Pseudo Gloss

Although the provided text-gloss example pairs may inherently reflect some structural characteristics of gloss annotation, the pseudo glosses generated by the LLM tend to follow the word order of the spoken language. As a result, they often fail to align properly with the sign sequence in the corresponding video. Consequently, supervision approaches such as CTC loss, which rely on strict sequential alignment, are not suitable for training with current pseudo glosses.

To address this issue, we need to correct the order of pseudo glosses. Currently, the pseudo glosses are generated purely from text, lacking any temporal grounding from the video itself. However, the video inherently contains rich temporal cues that can guide the correct sequencing of signs. Given the LLM Gloss $\tilde{\mathcal{G}}_{\mathcal{V}^i} = \{\tilde{g}_i, \tilde{g}_j, ..., \tilde{g}_l\}$ generated in Sec 3.2, we propose a video-guided refinement process in a weakly supervised manner to better align $\tilde{\mathcal{G}}_{\mathcal{V}^i}$ with the natural temporal structure of the video. To achieve this, we first denote a video-specific vocabulary set as $\tilde{\mathcal{U}}_{\mathcal{V}^i} = \{\tilde{g}_1, ..., \tilde{g}_n\}$, where $n$ is the number of unique glosses for video $\mathcal{V}^i$. Aggregating the pseudo glosses across the entire dataset, we construct a global vocabulary set denoted as $\tilde{\mathcal{U}} = \{\tilde{g}_1, ..., \tilde{g}_K\}$ of size $K$, representing all unique glosses generated by the LLM. Then, we extract frame-wise visual features from every video $\mathcal{V}^i$ using a vision encoder $\psi_V$, yielding a feature sequence $\mathbf{F}^i = (\mathbf{f}_1, ..., \mathbf{f}_T) \in \mathbb{R}^{T \times D}$, where $D$ is the feature dimension. We then use a classifier $\phi_{cls}$ to map these features to the global pseudo gloss vocabulary $\tilde{\mathcal{U}}$, producing a sequence of logits $\mathbf{Z}^i \in \mathbb{R}^{T \times K}$. Since each frame should correspond to at most one gloss label, we apply a softmax activation along the vocabulary dimension ($K$):

$$\mathbf{P} = \text{softmax}(\mathbf{Z}), \tag{5}$$

where $\mathbf{P}$ represents the frame-wise probability distribution over the global pseudo gloss vocabulary.

***Weakly Supervised Multi-Label Classification.*** Here, we describe the training process for the classifier $\phi_{cls}$. Since we only have weak supervision, *i.e.*, we have an estimated set (by LLM) of which words ($\tilde{\mathcal{U}}_{\mathcal{V}^i}$) are present in the video but lack precise timestamp annotations, we frame this as a multi-label classification (MLC) problem [40]. During the training phase, we first consolidate the predictions $\mathbf{P} \in \mathbb{R}^{T \times K}$ into a video-level representation by applying a max-pooling operation over the temporal dimension ($T$):

$$\hat{\mathbf{y}}_k = \max_{t \in [1,T]} \mathbf{P}_{t,k}, \quad \forall k \in \{1, ..., K\}, \tag{6}$$

where $\hat{\mathbf{y}} \in \mathbb{R}^K$ denotes the predicted presence probabilities for each gloss in the global vocabulary. We then optimize the classifier using Binary Cross-Entropy (BCE) loss:

$$\mathcal{L}_{\text{BCE}} = -\sum_{k=1}^{K} w_k \left[ y_k \log \hat{y}_k + (1 - y_k) \log(1 - \hat{y}_k) \right] \tag{7}$$

where $y_k \in \{0, 1\}$ denotes whether gloss $k$ appears in the draft (LLM-generated) pseudo gloss, and $w_k$ is a weighting factor. To address the long-tailed distribution of pseudo gloss words—where only a small subset of the vocabulary appears in each video (e.g., 7 vs. $1,100$ in Phoenix14T [8])—we introduce frequency-aware weighting to reweight the BCE loss:

$$w_k = w_{\text{base}} + \log \left( f_{\max} / f_k \right) \tag{8}$$

where $f_k$ is the frequency of gloss $k$, and $f_{\max}$ is the frequency of the most common gloss. These frequencies are computed based on the initial pseudo gloss drafts. This encourages the model to pay more attention to rare glosses, mitigating the imbalance problem.

***Temporal Smoothing Constrain.*** To enforce temporal consistency and encourage smooth transitions between consecutive frames, we introduce an L1 regularization loss on the frame-wise logits:

$$\mathcal{L}_{\text{smooth}} = \frac{1}{T - 1} \sum_{t=1}^{T-1} \|\mathbf{P}_t - \mathbf{P}_{t+1}\|_1 \tag{9}$$

This penalizes abrupt changes in gloss predictions across adjacent frames. The final loss function combines the BCE loss for weakly supervised gloss prediction and the L1 smoothness loss:

$$\mathcal{L} = \mathcal{L}_{\text{BCE}} + \mathcal{L}_{\text{smooth}} \tag{10}$$

***Frame-wise Gloss.*** Based on the frame-wise probability distribution $\mathbf{P} \in \mathbb{R}^{T \times K}$, we can perform a max-pooling operation along the vocabulary dimension ($K$) to derive the frame-wise gloss $\hat{\mathbf{y}}_{\text{gloss}} = \max_{k \in [1, K]} \mathbf{P}_{t,k} \in \mathbb{R}^T$.

***Greedy Algorithm for Reordering LLM Gloss.*** With the frame-wise probability distribution $\mathbf{P}_t$ obtained from Eq. 5, our goal is to refine the LLM Gloss to better align with $\mathbf{P}_t$ by designing a greedy algorithm that iteratively adjusts the LLM Gloss to resemble the frame-wise predictions. The process involves the following steps:

- *(i) Filter irrelevant words.* We first filter out words from the Frame-wise Gloss that are not present in the **LLM Gloss**, ensuring that only relevant terms are considered.

- *(ii) Merge consecutive duplicates.* Since certain glosses may be predicted for multiple consecutive frames, we consolidate adjacent duplicates to form a streamlined **Ref Gloss**.

- *(iii) Greedy reordering.* We perform a greedy reordering strategy to better align the LLM Gloss with the temporal structure suggested by the Ref Gloss. The key intuition is to leverage the order of words in the Ref Gloss as a soft temporal reference to reorganize the LLM Gloss, resulting in a refined sequence denoted as $\tilde{\mathcal{G}}_{\mathcal{V},target}$. If a word in the LLM Gloss is absent from the Ref Gloss, it is directly retained in its original order. For words that appear in both glosses but are out of order, we reorder them according to their first appearance in the Ref Gloss. More details are shown in Sec. A.2 in the appendix. This adaptive alignment ensures that the final sequence respects the temporal cues inferred from the video, enhancing consistency and coherence with the visual content.

Table 3 and Algorithm 1 provide a illustration of the reordering process. Once reordered, the resulting $\tilde{\mathcal{G}}_{\mathcal{V},\text{target}}$ is used to train the SLT model, following the strategy described in Sec. 3.4.

## 3.4 Training and Inference

Our training pipeline is illustrated in Figure 1. After generating the reordered gloss, we decompose the network training into three stages: *i.e.*, *Sign2Gloss*, *Gloss2Text*, and *Sign2Text*, following prior gloss-based approaches [10, 11] discussed in Sec. 3.1. In stage 1 (Sign2Gloss), we train a vision encoder to map sign video features to pseudo gloss sequences using the CTC loss. In stage 2 (Gloss2Text), we leverage a pre-trained multilingual language model, such as mBART [41] or Gemma2 [50], and fine-tune it on the gloss-to-text translation task. In stage 3 (Sign2Text), we introduce an additional projection layer, implemented as an MLP with two hidden layers randomly initialized.

| Text ($\mathcal{S}$) | im norden und westen mal sonne mal wolken (In the north and west sometimes sun sometimes clouds.) |
|---|---|
| **True Gloss** ($\mathcal{G}$) | NORDWEST / WECHSELHAFT / SONNE / WOLKE (northwest / changeable / sun / cloud) |
| LLM Gloss ($\tilde{\mathcal{G}}_{\mathcal{V}}$) | NORD / WEST / SONNE / WOLKE / WECHSELHAFT (north / west / sun / cloud / changeable) |
| Frame-wise Gloss | NORD / `NORD` / WEST / WECHSELHAFT / `WECHSELHAFT` / SONNE / `SONNE` / WOLKE / `WOLKE / WOLKE` / `REGION` |
| Ref Gloss | NORD / WEST / WECHSELHAFT / SONNE / WOLKE |
| **Reordered Gloss** ($\tilde{\mathcal{G}}_{\mathcal{V},target}$) | NORD / WEST / WECHSELHAFT / SONNE / WOLKE (north / west / changeable / sun / cloud) |

Table 3: Example of the LLM-generated pseudo gloss (LLM Gloss) and corresponding reorder operation. `NORD` means merging consecutive duplicates. `REGION` means discarding words not present in the LLM Gloss.

The pseudo gloss serves as an auxiliary supervision signal via the CTC loss, similar to stage 1. However, since the pseudo gloss is not an exact transcription compared to the true gloss, we remove the CTC loss in the final epochs (Stage 3.2) of training to prevent over-reliance on its potentially noisy supervision.

At test time, the model takes video frames as input and directly generates spoken language text, bypassing the intermediate pseudo gloss. This approach allows the model to

---

**Algorithm 1:** Reordering of LLM Gloss

**Input:** LLM Gloss: $\mathcal{L} = [l_1, ..., l_m]$, Ref Gloss: $\mathcal{R} = [r_1, ..., r_n]$
**Output:** Reordered Gloss: $target = [t_1, ..., t_m]$

1 Initialize $target \leftarrow [\,]$ (empty sequence of length $|\mathcal{L}|$);
2 Initialize two pointers: $i \leftarrow 0, j \leftarrow 0$;
3 **while** $i < |\mathcal{L}|$ **do**
4      **if** $l_i = r_j$ **then**
5          Append $l_i$ to $target$;
6          $i \leftarrow i + 1, j \leftarrow j + 1$;
7      **else if** $l_i \notin \mathcal{R}$ **then**
8          Append $l_i$ to $target$;
9          $i \leftarrow i + 1$;
10      **else if** $r_j$ appears at a later position in $\mathcal{L}$ (denoted as $l_{j'}$) **then**
11          Append $r_j$ to $target$;
12          Remove $l_{j'}$ from $\mathcal{L}$;
13          $j \leftarrow j + 1$;
14 **return** $target$

---

leverage structured pseudo gloss supervision during training to bridge vision and language, while maintaining the flexibility of end-to-end inference for fluent and accurate text generation.

## 4 Experiments

### 4.1 Experimental Setup

**Datasets.** We evaluate our proposed method on two widely used SLT datasets. Phoenix14T [8] consists of 8,257 fully glossed German Sign Language (DGS) videos with corresponding German translations and sign glosses, sourced from weather forecast programs. The dataset is split into train/dev/test sets, containing 7,096/519/642 videos, respectively. How2Sign [16] is a large-scale American Sign Language (ASL) dataset without gloss annotation, featuring 79 hours of videos with English translations, containing a vocabulary of approximately 16,000 unique English words. We utilize the manually re-aligned[4] video clips, with train/dev/and test splits consisting of 31,101/1,527/2,349 samples, respectively.

**Evaluation Metrics.** Following prior works [10, 12, 28], we adopt BLEU [42] and ROUGE [39] to assess SLT performance. Higher BLEU and ROUGE scores indicate better translation quality.

**Implementation Details.** Following [11, 10], we adopt S3D [54], pre-trained on Kinetics-400 [31], as our visual encoder for sign videos. Each input video of shape $T \times H \times W \times 3$ is fed into the encoder, where $T$ denotes the number of frames, and $H$ and $W$ represent the height and width of the video, respectively. The encoder extracts spatiotemporal features, which are then spatially pooled to a size of $T/4 \times 512$. These features are passed through a projector consisting of two fully connected layers. For the translator, we initialize either mBART[2] [41] or Gemma2[3] [50] with pre-trained checkpoints from HuggingFace [52]. We use AdamW optimizer along with a cosine annealing learning rate schedule. The total training process is divided into four stages, with 40 epochs allocated to stages 1, 2, 3.1, and 3.2, respectively. More implementation details are reported in Appendix.

---

[4]https://how2sign.github.io/, [2]https://huggingface.co/facebook/mbart-large-cc25, [3]https://huggingface.co/google/gemma-2-2b-it

## 4.2 Comparison against State-of-the-art Methods

**Results on Phoenix14T**. We present our main results in Table 4. Our proposed PGG-SLT significantly outperforms previous gloss-free methods. When using the same mBART [41] language decoder, PGG-SLT improves the BLEU4 score by 5.0/5.4 over GFSLT-VLP [59] and 0.5/0.8 over VAP [28] on the Dev/Test sets, respectively. Compared to the previous state-of-the-art VAP [28], our method does not require either a skeleton-specific encoder [29] or any additional training of a translation model. Surprisingly, PGG-SLT achieves performance competitive with gloss-based methods such as MMTLB [10], while using only 30 gloss annotations out of 7K available in the dataset. For any SLT dataset, the cost of annotating a few dozen glosses is negligible. This result suggests that our pseudo-glosses closely approximate real gloss annotations, significantly boosting performance compared to other pseudo-gloss-based approaches like Sign2GPT [53]. Furthermore, when equipped with a stronger LLM translator such as Gemma2-2B [50], PGG-SLT achieves even better results—surpassing gloss-based methods on metrics such as BLEU1 score. This improvement can likely be attributed to Gemma2's richer vocabulary (tokenizer), enabling more precise and diverse language generation.

**Results on How2Sign**. We further evaluate our method on the more challenging How2Sign [16] dataset (more details in Sec A.1 in appendix), with results summarized in Table 5. Since How2Sign does not provide gloss annotations, gloss-based methods are not applicable in this setting. Compared to competitor methods, our models consistently outperform VAP [28] in both BLEU4 and ROUGE scores. When compared to SSVP-SLT [45], which leverages self-supervised learning to pre-train the visual encoder, our best model achieves a 6.1 BLEU4 improvement on the test set. While prior works [51, 38, 45] have demonstrated that large-scale training benefits SLT models, our approach remains highly competitive—even surpassing Uthus *et al.* [51] by a significant margin. These results highlight the effectiveness of our pseudo gloss generation in facilitating better alignment and training of SLT models. Importantly, the strong performance on both Phoenix14T [8] and How2Sign [16] demonstrates that LLMs can serve as powerful assistants for generating high-quality pseudo gloss annotations. Furthermore, our method scales effectively with dataset size, making it well-suited for real-world, large-scale SLT applications.

| Method | Dev Set | | | | | Test Set | | | | |
|---|---|---|---|---|---|---|---|---|---|---|
| | BLEU1 | BLEU2 | BLEU3 | BLEU4 | ROUGE | BLEU1 | BLEU2 | BLEU3 | BLEU4 | ROUGE |
| *Gloss-based* | | | | | | | | | | |
| MMTLB [10] | 53.95 | 41.12 | 33.14 | 27.61 | 53.10 | 53.97 | 41.75 | 33.84 | 28.39 | 52.65 |
| SLTUNet [57] | - | - | - | 27.87 | 52.23 | 52.92 | 41.76 | 33.99 | 28.47 | 52.11 |
| IP-SLT [55] | 54.10 | 41.56 | 33.66 | 28.22 | 54.43 | 54.25 | 41.51 | 33.45 | 27.97 | 53.72 |
| TS-SLT [11] | 54.32 | 41.99 | 34.15 | 28.66 | 54.08 | 54.90 | 42.43 | 34.46 | 28.95 | 53.48 |
| *Gloss-free* | | | | | | | | | | |
| NSLT [8] | 31.58 | 18.98 | 13.22 | 10.00 | 32.60 | 29.86 | 17.52 | 11.96 | 9.00 | 30.70 |
| GASLT [56] | - | - | - | - | - | 39.07 | 26.74 | 21.86 | 15.74 | 39.86 |
| GFSLT-VLP [59] | 44.08 | 33.56 | 26.74 | 22.12 | 43.72 | 43.71 | 33.18 | 26.11 | 21.44 | 42.49 |
| SignLLM [18] | 46.88 | 36.59 | 29.91 | 25.25 | 47.23 | 45.21 | 34.78 | 28.05 | 23.40 | 44.49 |
| Sign2GPT [53] | - | - | - | - | - | 49.54 | 35.96 | 28.83 | 22.52 | 48.90 |
| FLa-LLM [12] | - | - | - | - | - | 46.29 | 35.33 | 28.03 | 23.09 | 45.27 |
| VAP [28] | 52.78 | - | - | 26.62 | 51.47 | 53.07 | - | - | 26.02 | 51.28 |
| PGG-SLT (mBART) | 53.16 | 40.38 | 32.66 | 27.09 | 52.01 | 53.45 | 40.55 | 32.65 | 26.85 | 51.85 |
| PGG-SLT (Gemma2) | 53.84 | 41.08 | 32.82 | 27.53 | 52.85 | 54.02 | 41.23 | 32.89 | 27.32 | 52.56 |

Table 4: Comparison with state-of-the-art methods on Phoenix14T. We report results using two translators: mBART [41] and Gemma2-2B [50]. Equipped with generated pseudo gloss, our approach surpasses previous gloss-free methods and achieves performance comparable to gloss-based MMTLB.

## 4.3 Ablation Study

**In-context Learning Ability of LLM.** Our key hypothesis is that LLMs can act as human experts to extract key lexical information from a sentence and generate corresponding gloss annotations. Although sign language differs systematically from spoken language, *e.g.*, in grammar or the use of named entities, these modality-specific biases may not be immediately captured by LLMs. However, such gaps can be narrowed down by providing LLM with a small number of gloss-text example pairs, allowing it to adapt to the unique linguistic patterns of sign language through in-context learning. As shown in Table 6a, even without any examples, LLM-generated glosses achieve a BLEU4 score

| Method | Dev Set | | | | | Test Set | | | | |
|---|---|---|---|---|---|---|---|---|---|---|
| | BLEU1 | BLEU2 | BLEU3 | BLEU4 | ROUGE | BLEU1 | BLEU2 | BLEU3 | BLEU4 | ROUGE |
| Uthus *et al.* [51] | - | - | - | - | - | 15.0 | 5.1 | 2.3 | 1.2 | - |
| SSVP-SLT [45] | - | - | - | - | - | 30.2 | 16.7 | 10.5 | 7.0 | 25.7 |
| Tarrés *et al.* [48] | 35.2 | 20.6 | 13.3 | 8.9 | - | 34.0 | 19.3 | 12.2 | 8.0 | - |
| FLa-LLM [12] | - | - | - | - | - | 29.8 | 19.0 | 13.3 | 9.7 | 27.8 |
| VAP [28] | 42.3 | - | - | 14.9 | 30.3 | 39.2 | - | - | 12.9 | 27.8 |
| Uthus *et al.*[†] [51] | - | - | - | - | - | 37.8 | 24.1 | 16.9 | 12.4 | - |
| Uni-Sign[†] [38] | - | - | - | - | - | 40.2 | - | - | 14.9 | 36.0 |
| SSVP-SLT[†] [45] | - | - | - | - | - | 43.2 | 28.8 | 20.8 | 15.5 | 38.4 |
| PGG-SLT (mBART) | 41.8 | 26.5 | 19.7 | 15.2 | 32.6 | 38.9 | 25.4 | 18.1 | 13.1 | 31.5 |
| PGG-SLT (Gemma2) | 43.4 | 27.7 | 21.1 | 15.9 | 34.8 | 40.8 | 25.9 | 18.6 | 13.7 | 32.9 |

Table 5: Comparison with SOTA methods on How2Sign. We report results using two translators: mBART [41] and Gemma2-2B [50]. [†] indicates the models use an extra large-scale dataset [46, 51] for pre-training.

| # Pairs | Gloss (Train) | | | Gloss (Test) | | |
|---|---|---|---|---|---|---|
| | WER$^{\downarrow}$ | B1$^{\uparrow}$ | B4$^{\uparrow}$ | WER$^{\downarrow}$ | B1$^{\uparrow}$ | B4$^{\uparrow}$ |
| POS [53] | 95.8 | 28.7 | 2.1 | 96.5 | 28.2 | 2.0 |
| 0 | 67.5 | 44.6 | 10.7 | 67.1 | 43.2 | 10.5 |
| 30 | 62.8 | 53.1 | 14.3 | 62.1 | 53.6 | 13.8 |
| 700 | 61.2 | 55.2 | 17.9 | 60.9 | 55.7 | 17.2 |

(a) Quality of generated pseudo gloss.

| Pseudo Gloss | Test set | |
|---|---|---|
| | B1$^{\uparrow}$ | B4$^{\uparrow}$ |
| POS [53] | 46.3 | 20.8 |
| # Pairs = 0 | 52.5 | 25.6 |
| # Pairs = 30 | 53.1 | 26.2 |
| True Gloss (7K) | 53.9 | 28.2 |

(b) SLT results.

| LLM | Gloss (Test) | | |
|---|---|---|---|
| | WER$^{\downarrow}$ | B1$^{\uparrow}$ | B4$^{\uparrow}$ |
| Gemma2-7B | 65.9 | 48.5 | 11.9 |
| Gemma2-27B | 65.2 | 49.2 | 12.2 |
| GeminiXL-IT | 63.9 | 52.8 | 12.6 |
| Gemini1.5Pro | 62.1 | 53.7 | 13.8 |

(c) Different LLMs (30 pairs).

| Reorder | Gloss (Train) | | | PHX [8] | | H2S [16] | |
|---|---|---|---|---|---|---|---|
| | WER$^{\downarrow}$ | B1$^{\uparrow}$ | B4$^{\uparrow}$ | B1$^{\uparrow}$ | B4$^{\uparrow}$ | B1$^{\uparrow}$ | B4$^{\uparrow}$ |
| ✗ | 62.8 | 53.1 | 14.3 | 53.1 | 26.0 | 38.5 | 12.6 |
| ✓ | 60.1 | 53.1 | 17.4 | 53.5 | 26.9 | 38.9 | 13.1 |

(d) Reorder operation for pseudo gloss (30 paris).

| $\mathcal{L}_{smooth}$ | $w_k$ | Gloss (Train) | | | | Gloss (Test) | | | |
|---|---|---|---|---|---|---|---|---|---|
| | | Prec.$^{\uparrow}$ | Rec.$^{\uparrow}$ | WER$^{\downarrow}$ | B4$^{\uparrow}$ | Prec.$^{\uparrow}$ | Rec.$^{\uparrow}$ | WER$^{\downarrow}$ | B4$^{\uparrow}$ |
| ✗ | ✗ | 86.7 | 99.3 | 30.2 | 55.4 | 73.0 | 83.5 | 43.5 | 36.3 |
| ✗ | ✓ | 86.2 | 99.1 | 30.0 | 53.5 | 73.2 | 84.8 | 42.9 | 36.1 |
| ✓ | ✗ | 90.7 | 99.2 | 25.6 | 57.6 | 76.6 | 84.1 | 37.0 | 41.4 |
| ✓ | ✓ | 91.7 | 99.5 | 22.8 | 61.9 | 78.1 | 85.2 | 36.1 | 43.1 |

(e) Components for training the reordering classifier.

Table 6: Ablation study on Phoenix14T: (a) the similarity between generated pseudo gloss and true gloss, measured by Word Error Rate (WER) and BLEU score, "POS" means the Parts-of-Speech tagging in [53]; (b) SLT performance using different pseudo gloss variants; (c) the similarity between pseudo gloss generated by different LLMs and the true gloss. Our default setting is shown by  gray .

of 10.7 when compared with ground-truth glosses, significantly outperforming prior heuristic-based pseudo gloss methods [53] (labelled "POS" in the table), with over 30% relative improvement in WER and an 8% increase in BLEU4. When provided with 30 example pairs, the BLEU4 score of generated glosses increases to 14%. The performance continues to improve as more examples are given. While there's still headroom for improvement, the progress so far is already significant, demonstrating the effectiveness of LLMs' in-context learning capabilities.

**Effectiveness of the Pseudo Gloss.** We present the SLT results trained (excluding stage 3.2) with different types of pseudo glosses in Table 6b. The glosses generated by the LLM already provide a strong baseline under a fully gloss-free setting. When 30 ground-truth gloss annotations are included as in-context examples, the final translation quality further improves. Using the full set of ground-truth glosses serves as the performance upper bound of our approach.

**Ablation on Different LLMs.** To isolate the impact of LLMs, we compare different model versions and parameter scales while using the same 30 pairs for long-context in-context learning. As shown in Table 6c, larger models, such as Gemma2-27B [50], tend to generate higher-quality glosses. More advanced models, such as Gemini 1.5 Pro [49], further improve the quality of generated glosses and achieve the best performance.

**Effectiveness of the Reordering Operation.** We evaluate the proposed reordering operation in Table 6d. The core improvement introduced by this operation lies in reducing the linguistic asymmetry between LLM-generated glosses, which follow spoken language syntax, and ground-truth sign language glosses, which are aligned with the visual signs in the video. After reordering, the similarity between the pseudo glosses and the ground-truth glosses is improved, leading to better performance during the training of the SLT model.

**Ablation on Weakly Supervised Classification.** We employ two constraints to assist in training the classifier for reordering (Sec 3.3): temporal smoothing constraint $\mathcal{L}_{smooth}$ and the frequency-aware weighting coefficient $w_k$. To evaluate their effectiveness, we conduct this ablation study using ground-truth glosses while excluding their temporal order information. As shown in Table 6e, the baseline MLC solution, using BCE loss, achieves satisfactory precision and recall. With the addition of $\mathcal{L}_{smooth}$, both WER and BLEU4 scores significantly improve, demonstrating that the predicted order is closer to the ground-truth. The $w_k$ further enhances the accuracy of the pseudo gloss.

**Effectiveness of Vision-Language Alignment in Training.** Our key contribution lies in generating high-quality pseudo glosses to enhance the training of SLT models. Figure 1 outlines the overall training pipeline. A key question is: *what happens if no gloss annotations are used*? As shown in Figure 2 and Table 7, directly training the *Sign2Text* model using only sentence-level supervision (Stage 3.2) results in poor performance—yielding BLEU4 scores of just 8.2 and 2.3 on PHOENIX14T and How2Sign, respectively. Increasing training epochs does not help and even degrades performance. In contrast, incorporating our pseudo glosses (via Stage 1 and Stage 2) leads to substantial performance gains, underscoring the critical role of vision-language alignment in SLT training. Moreover, higher-quality glosses yield stronger alignment signals, further boosting model performance.

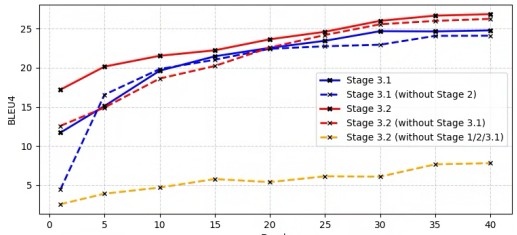

Figure 2: BLEU4 *vs.* epochs on Phoenix14T test set.

| Train Stage | | | | PHX [8] | | H2S [16] | |
|---|---|---|---|---|---|---|---|
| 1 | 2 | 3.1 | 3.2 | BLEU1 | BLEU4 | BLEU1 | BLEU4 |
| ✗ | ✗ | ✗ | ✓ | 24.2 | 8.2 | 17.5 | 2.3 |
| ✓ | ✓ | ✓ | ✗ | 50.9 | 24.8 | 37.2 | 11.5 |
| ✓ | ✓ | ✗ | ✓ | 52.3 | 26.2 | 38.2 | 12.7 |
| ✓ | ✓ | ✓ | ✓ | 53.5 | 26.9 | 38.9 | 13.1 |

Table 7: Impact of different training stages.

## 5   Conclusion

In this paper, we propose leveraging LLMs to address the challenging problem of Sign Language Translation (SLT) by generating pseudo glosses in a gloss-free setting. Our method, PGG-SLT, demonstrates significant performance improvements on the Phoenix14T and How2Sign datasets. We show that LLMs can serve as experts in extracting key information from sentences, and their in-context learning capabilities help capture potential grammatical patterns and glossing conventions. Moreover, we introduce a weakly supervised setting to reorder the generated pseudo glosses, mitigating the misalignment between LLM-generated glosses and the sign sequences in the video. We believe this approach presents a promising direction for leveraging LLMs to enhance SLT model training.

**Limitations and Future Work.** One limitation of this work is that we only scaled the dataset from Phoenix14T to How2Sign, and the performance of our proposed PGG-SLT when incorporating additional training data remains unvalidated. As shown in Table 5, two other methods that leverage large-scale datasets for pre-training achieve slightly better results than our model. Moving forward, we plan to evaluate the effectiveness of our pseudo-gloss generation method on larger ASL benchmarks. In addition, this paper is mainly about pseudo gloss generation, we first want to clarify how *glossing* are defined and described here. Indeed, reducing certain action video frames to a few "words" inherently limits the expressiveness of sign language in such representations. This is why resources like 2M-Flores-ASL incorporate markers such as "+++" to indicate repeated actions, "cl:" to denote specific handshape classifiers, as well as markers like "//WHAT\\" and "//WHY\\" to represent NMMs (with // and \\ indicating their span) in their dataset. Further, sign language exhibits substantial variability, different sign sequences can convey the same meaning, much like how spoken English uses diverse vocabulary for identical concepts. Crucially, text (spoken language) alone cannot perfectly reverse-engineer the specific visual details (*e.g.*, precise hand movements, spatial configurations) captured in video. That said, given current constraints (*e.g.*, dataset scale and model capabilities), we think that our current relatively "simple" glossing approach offers practical advantages: it lowers the learning barrier for models and facilitates more measurable improvements in SLT performance (*e.g.*, BLEU scores). Implement a more nuanced, expressive representation will also be our future work.

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

# Appendix

## A Training Details

### A.1 Dataset Preparation

> "role": "user",    "content": "<Example: [Text, Gloss], ..., [Text, Gloss]>, <Instruction>, **<Query Text>**"
>
> "role": "assistant",    "content": "**<LLM-Generated Pseudo Gloss>**"

Figure 3: Example prompt used for pseudo gloss generation with Gemini 1.5 Pro [49]. The prompt contains a few example text-gloss pairs to guide the LLM in generating well-structured glosses for the query text. Detailed prompt formatting can be found in Figure 4 and Figure 6.

**Phoenix14T.** Figure 4 presents examples of prompts used to generate pseudo glosses with an LLM. The content under the user role serves as the input to the LLM, while the content under the assistant role represents the LLM's output.

In practice, we randomly select 30 true glosses from the Phoenix14T training set to create example text-gloss pairs. The quantitative results are presented in Table 6a of the main paper. Figure 5 illustrates the prompt without text-gloss pairs, corresponding to the second row of Table 6a.

```
# Prompt (w/ 2 example pairs) and LLM's answer (shown in bold)

LLM Input:
{
    "role": "user",
    "content": f"""
                German text to German gloss examples:

                Example 1: text: [und nun die wettervorhersage für morgen donnerstag den zwölften august] gloss: [JETZT
                WETTER MORGEN DONNERSTAG ZWOELF FEBRUAR]
                Example 2: text: [mancherorts regnet es auch länger und ergiebig auch lokale überschwemmungen sind wieder
                möglich] gloss: [ORT REGEN DURCH REGEN KOENNEN UEBERSCHWEMMUNG KOENNEN]

                These examples demonstrate the transformation from natural German text into structured German gloss, specifically
                based on the benchmark for sign language translation in German weather forecasts.

                The output gloss should be optimized for sign language video translation by considering which words (e.g., nouns,
                adjectives, adverbs, numbers) are crucial in the context of weather forecasting.

                The gloss should:
                - Exclude function words like articles ("der, die, das") and auxiliary verbs unless necessary.
                - Preserve key content words such as weather conditions, time expressions, and geographic references.
                - Follow the typical structure of German sign language glosses for video.

                Now, based on the examples provided above, please infer the most suitable German gloss sequence for the following
                German text:

                text: [ab donnerstag breiten sich regenwolken allmählich weiter richtung südosten aus]

                The output should follow the following format without any text formatting:
                Output: the generated gloss."""
}
LLM Output:
{
    "role": "assistant",
    "content": "[DONNERSTAG REGEN WOLKE LANGSAM KOMMEN SUEDOST]"
},
```

Figure 4: The prompt sent to the LLM for generating pseudo glosses includes two example pairs. These two examples serve as references to guide the LLM in producing accurate pseudo glosses during generation. The text marked in red represents spoken language from the datasets, which should be replaced during each iteration.

**How2Sign.** How2Sign [16] is an ASL benchmark without gloss annotations, we randomly sample 20 glosses from the 2M-Flores-ASL dataset[5] to form the example pairs. The constructed prompt and representative text-gloss pairs for How2Sign are illustrated in Figure 6.

Table 8 presents examples of the final pseudo glosses for the How2Sign dataset. In cases where the generated and reordered gloss annotations exceed the total sequential length of the video—making the application of CTC loss infeasible—we remove redundant words from the generated glosses. The list of such words is provided below.

```
# Prompt (w/o example pairs) and LLM's answer (shown in bold)

LLM Input:
{
    "role": "user",
    "content": f"""Your task is to generate German Sign Language gloss based on the given German text. Your gloss generation
                should align with the standard glossing conventions used in German Sign Language, particularly in the context of
                weather forecasting. The output gloss should be optimized for sign language video translation by considering which
                words (e.g., nouns, adjectives, adverbs, numbers) are crucial in the context of weather forecasting.

                The gloss should:
                - All numbers must be written in German words (e.g., "eins", "zwei", "drei").
                - Exclude function words like articles ("der, die, das") and auxiliary verbs unless necessary.
                - Preserve key content words such as weather conditions, time expressions, and geographic references.
                - Follow the typical structure of German sign language glosses for video.

                text: [ab donnerstag breiten sich regenwolken allmählich weiter richtung südosten aus]

                The output should follow the following format without any text formatting:
                Output: the generated gloss."""
}

LLM Output:
{
    "role": "assistant",
    "content": "[DONNERSTAG REGENWOLKEN ALLMÄHLICH SÜDOSTEN AUSBREITEN]"
},
```

Figure 5: The prompt sent to the LLM for generating pseudo glosses without providing example pairs. The text marked in red represents spoken language from the datasets, which should be replaced during each iteration.

**Discussion on choosing text-gloss pairs.** In this paper, we directly use random selection to choose the text-gloss pairs. We conducted experiments comparing text-gloss vs. gloss-text pair formats in prompt, e.g., "Example1: text: [] gloss: []" (as shown in Figure 4) vs. "Example 1: gloss: [] text: []". For generated gloss quality, using text-gloss pairs as prompts yields pseudo-glosses closer to true glosses (with better BLEU1 and BLEU4 scores) than gloss-text prompts, showing that prompt engineering is a critical factor in optimizing performance. We leave other selecting approaches for future study; selecting better text-gloss example pairs may ensure greater diversity and reduce bias.

## A.2   Greedy Reordering Strategy

Here, we detail our Greedy Reordering Algorithm for LLM Gloss, as outlined in Sec. 3.3 and illustrated in Algorithm 1. This strategy aims to optimally align the LLM Gloss with the temporal structure suggested by the Ref Gloss. The process starts by initializing an empty sequence, denoted as *target*, with the same length as the LLM Gloss. Two pointers, $i$ and $j$, are set to 0, representing the current positions in LLM Gloss and Ref Gloss, respectively. At each step, we compare the current words from both sequences: $l_i$ from the LLM Gloss and $r_j$ from the Ref Gloss. If $l_i = r_j$, we append $l_i$ to $target$ and increment both pointers ($i \leftarrow i+1, j \leftarrow j+1$). If $l_i \neq r_j$ and $l_i$ does not exist in the reference gloss, we directly append $l_i$ to $target$ and move the LLM pointer forward ($i \leftarrow i+1$). In the final case, if $l_i \neq r_j$ but $r_j$ is found at a later position in the LLM Gloss, denoted as $l_{j'}$, we append $r_j$ to $target$, remove the occurrence of $l_{j'}$ from the LLM Gloss, and increment the reference pointer ($j \leftarrow j+1$). This process continues until all elements in the LLM Gloss are processed, resulting in a reordered version that aligns more accurately with the temporal structure suggested by Ref Gloss.

---

[5] https://huggingface.co/datasets/facebook/2M-Flores-ASL

Figure 6: Example prompt that we feed to Gemini 1.5 Pro [49] for pseudo gloss generation on How2Sign [16].

| | |
|---|---|
| **Text** | this is dr art bowler and this has been how to tell if you have low self esteem |
| Gloss | doctor a r t b o w l e r how tell self esteem low |
| **Text** | so i you know i had to just fight through the cleaning and had to have a lock smith come out and open my doors and i had to go to another job |
| Gloss | so ix know ix must fight through clean must locksmith come open door ix must another job go |
| **Text** | again we're talking about cross stitch cross stitch craft kits |
| Gloss | repeat discuss cross stitchcross stitch craft kit |
| **Text** | i'm cooie grey lavin and in this clip i'll show you how to use unique headstones to customize your yard |
| Gloss | me c o o i e g r e y l a v i n this clip show how use unique headstone customize yard |
| **Text** | what she's displaying is the angry colors |
| Gloss | she show angry color |
| **Text** | try it with a little tempo |
| Gloss | ix try little fast |
| **Text** | and most the time you can see that if you look hopefully you can pick this up here but you'll see high speed it'll usually be stamped right on it |
| Gloss | time can see look hope you see high speed usually stamp right there |

Table 8: Generated pseudo glosses for How2Sign (punctuation marks are omitted).

Remove words = { "the", "a", "an", "on", "in", "to", "of", "at", "by", "this", "that", "and", "but", "or", "so", "because", "for", "with", "about", "as", "if", "then", "just", "like", "very", "really", "actually", "alright", "well", "now", "there", "here", "we", "you", "my", "your", "our", "us", "it", "its", "they", "them", "their", "is", "are", "was", "were", "be", "been", "being", "do", "does", "did", "done", "can", "could", "will", "would", "shall", "should", "might", "must", "let", "make", "get" }

Table 9 summarizes the number of generated pseudo glosses based on the approach detailed in Sec. 3.

| Dataset | # Vocab | # Pseudo Gloss $\tilde{\mathcal{G}}$ |
|---|---|---|
| Phoenix14T | 2,887 | 1,120 |
| How2Sign | 15,541 | 9,796 |

Table 9: Vocabulary size of text *vs.* pseudo glosses.

## A.3 Architecture Overview

Our framework follows the gloss-based architecture MMTL [10]. The visual encoder $\phi_V$ consists of an S3D backbone [54] and a head network. Given an input video $\mathcal{V} \in \mathbb{R}^{T \times 224 \times 224 \times 3}$, only the first four blocks of S3D are used, producing features $\mathbf{Z} \in \mathbb{R}^{T/4 \times 832}$ after spatial pooling. These features are then processed by a head network, comprising a projection block (temporal linear layer, BN, and ReLU), followed by a temporal convolutional block (two convolutional layers with kernel size 3, a linear layer, and ReLU). The final output is in the shape of $\mathbb{R}^{T/4 \times 512}$. In Stage 1 and Stage 3.1, a linear classifier and Softmax are applied to $\mathbf{Z}$, resulting in frame-level gloss probabilities $\mathbf{P} \in \mathbb{R}^{T/4 \times K}$ where $K$ denotes the gloss vocabulary size. In Stage 3.1, we introduce an additional randomly initialized projector, consisting of two fully connected layers. For the translation network (referred to as the "Translator" in Figure 1), we adopt mBART [41], a sequence-to-sequence denoising autoencoder, and Gemma-2 [50], a decoder-only architecture. Both models are initialized with pretrained weights from large-scale multilingual corpora, providing a strong backbone for our translation process. Table 10 provides a comprehensive breakdown of the number of trainable parameters for each component of our SLT model.

| Component | # Params | # Trainable |
|---|---|---|
| Visual Encoder | 4,734,272 | 4,695,936 |
| Visual Head | 8,280,906 | 8,280,906 |
| Projector | 1,574,912 | 1,574,912 |
| Translater (mBART) | 358,461,888 | 358,461,888 |
| Total | | 373,013,642 |

(a) Model parameters of mBART based translator.

| Component | # Params | # Trainable |
|---|---|---|
| Visual Encoder | 4,734,272 | 4,695,936 |
| Visual Head | 8,276,868 | 8,276,868 |
| Projector | 6,492,672 | 6,492,672 |
| Translater (Gemma2) | 2,045,709,312 | 12,779,520 |
| Total | | 32,244,996 |

(b) Model parameters of Gemma2-2B based translator.

Table 10: Training settings including (a) the number of pseudo-glosses during pre-training and (b) parameter counts during downstream translation.

## A.4 Training Configurations

We report our training configurations for our PGG-SLT in Table 11.

| Parameter | Stage 1 | Stage 2 | Stage 3.1 | Stage 3.2 |
|---|---|---|---|---|
| Model | S3D | mBART | S3D & mBART | S3D & mBART |
| Attention drop | - | 0.1 | 0.1 | 0.1 |
| Dropout | - | 0.3 | 0.3 | 0.3 |
| Label smoothing | - | 0.2 | 0.2 | 0.2 |
| Number of beams | - | 5 | 5 | 5 |
| Video size (T, C, H, W) | (T, 3, 224, 224) | - | (T, 3, 224, 224) | (T, 3, 224, 224) |
| Horizontal Flip | ✗ | ✗ | ✓ | ✓ |
| Temporal Sampling Range | [0.5, 1.5] | [0.5, 1.5] | [0.7, 1.3] | [0.7, 1.3] |
| Optimizer | AdamW | AdamW | AdamW | AdamW |
| Optimizer momentum ($\beta_1/\beta_2$) | 0.9/0.999 | 0.9/0.999 | 0.9/0.999 | 0.9/0.999 |
| Learning rate schedule | Cosine | Cosine | Cosine | Cosine |
| Weight decay | 0.05 | 0.05 | 0.05 | 0.05 |
| Warmup epochs | 5 | 5 | 0 | 0 |
| Epochs | 40 | 40 | 40 | 40 |
| Batch size | 8 | 8 | 8 | 8 |
| Peak learning rate | $5 \times 10^{-3}$ | $5 \times 10^{-5}$ | $1 \times 10^{-4}$ | $1 \times 10^{-5}$ |
| Minimum learning rate | $1 \times 10^{-7}$ | $1 \times 10^{-7}$ | $1 \times 10^{-7}$ | $1 \times 10^{-7}$ |

Table 11: Training settings on Phoenix14T.

## A.5 Connectionist Temporal Classification (CTC) Loss Formulation

The goal of the *Sign2Gloss* stage, also known as Sign Language Recognition (SLR), is to predict the gloss sequence $\mathcal{G} = (g_1, ..., g_M)$ from the sign video $\mathcal{V}$ containing $T$ frames. We employ the CTC loss [19], which allows for sequence-level supervision without requiring precise frame-to-gloss alignments. Specifically, CTC computes the probability of $\mathcal{G}$ by summing over all feasible

frame-to-gloss alignments:

$$p(\mathcal{G}|\mathcal{V}) = \sum_{\pi \in S} p(\pi|\mathcal{V}), \tag{11}$$

where $\pi$ denotes a frame-level gloss path, and $S$ is the set of all valid mappings between the visual frames and the gloss sequence. The probability $p(\pi|\mathcal{V})$ is obtained by applying a Softmax activation to the output of the visual encoder $\psi_V$. The CTC loss is then defined as the negative log-likelihood:

$$\mathcal{L}_{\text{SLR}} = -\log p(\mathcal{G}|\mathcal{V}; \psi_V). \tag{12}$$

In the *Gloss2Text* stage, the recognized gloss sequence $\mathcal{G}$ is translated into the target spoken language sentence $\mathcal{S} = (s_1, ..., s_U)$ using a neural translation model $\psi_T$. This is achieved through a sequence-to-sequence learning objective:

$$\mathcal{L}_{\text{SLT}} = -\sum_{i=1}^{U} \log p(s_i|s_{<i}, \mathcal{G}; \psi_T), \tag{13}$$

where $p(s_i|s_{<i}, \mathcal{G})$ represents the probability of generating the $i$-th word given its preceding words and the gloss sequence.

### A.6 More Results on CSL-Daily.

| Method | Dev Set | | | Test Set | | |
|---|---|---|---|---|---|---|
| | BLEU1 | BLEU4 | ROUGE | BLEU1 | BLEU4 | ROUGE |
| TS-SLT [11] | 58.24 | 29.18 | 57.81 | 58.64 | 29.55 | 58.62 |
| GFSLT-VLP* [59] | 39.20 | 11.07 | 36.70 | 39.37 | 11.00 | 36.44 |
| Sign2GPT* [53] | - | - | - | 41.75 | 15.40 | 42.36 |
| SignLLM* [18] | 42.45 | 12.23 | 39.18 | 39.55 | 15.75 | 39.91 |
| VAP* [28] | 53.31 | 23.84 | 51.19 | 52.98 | 23.65 | 51.09 |
| PGG-SLT* (mBART) | 53.58 | 24.05 | 52.34 | 53.29 | 23.70 | 52.88 |

Table 12: Comparison with SOTA methods on CSL-Daily [60]. * indicates the Gloss-free method.

### A.7 Qualitative Results

Table 13 and Table 16 provide qualitative examples of our mBART-based PGG-SLT model on the Phoenix14T and How2Sign datasets, respectively. The reordering of gloss annotations is illustrated in Table 14. We also compare our approach with the best-performing models, including Tarr'es *et al.* [48], Uthus *et al.* [51], SSVP-SLT [45], and the reference translations on How2Sign. As shown in Table15, although our method achieves a slightly lower BLEU4 score compared to SSVP-SLT, the generated predictions remain highly relevant and on-topic, comparable to SSVP-SLT. Notably, all current models still face challenges with repetitions and sign mix-ups in various instances.

| | | |
|---|---|---|
| Val Set (1) | **Reference** | sonst regnet es teilweise kräftig. (Otherwise it rains heavily at times.) |
| | Prediction | sonst regnet es teilweise kräftig. (Otherwise it rains heavily at times.) |
| Val Set (2) | **Reference** | am tag wechseln sonne und wolken einander ab teilweise ist es auch längere zeit sonnig. (During the day sun and clouds alternate with each other at times it is also sunny for a longer period of time.) |
| | Prediction | am tag sonne und wolken im wechsel gebietsweise zeigt sich die sonne für längere zei. (During the day sun and clouds alternate in some areas the sun appears for a longer period of time.) |
| Val Set (3) | **Reference** | das hoch bringt uns bis mindestens karfreitag überwiegend freundliches wetter und steigende temperaturen. (The high-pressure system brings us predominantly pleasant weather and rising temperatures until at least Good Friday) |
| | Prediction | bis freitag wird es unter hochdruckeinfluss langsam wieder freundlicher und auch die temperaturen steigen wieder. (By Friday under high-pressure influence it will slowly become more pleasant again and the temperatures will also rise again.) |
| Test Set (4) | **Reference** | am donnerstag regen in der nordhälfte in der südhälfte mal sonne mal wolken ähnliches wetter dann auch am freitag. (On Thursday rain in the northern half in the southern half sometimes sun sometimes clouds similar weather then also on Friday.) |
| | Prediction | am donnerstag regnet es an den küsten sonst sonne und wolken im wechsel am freitag wechselhaftes wetter. (On Thursday it rains on the coasts otherwise sun and clouds alternate on Friday changeable weather.) |
| Test Set (5) | **Reference** | teilweise ist es auch klar. (Partly it is also clear.) |
| | Prediction | teilweise klart es auch auf. (Partly it clears up as well.) |

Table 13: Phoenix14T qualitative results.

| (1) | **Text** ($\mathcal{S}$) | gegen abend zieht von westen wieder schnee heran der sich am sonntag weiter ausbreitet. (Toward evening snow moves in again from the west which will spread further on Sunday.) |
|---|---|---|
| | **True Gloss** ($\mathcal{G}$) | IX / ABEND / KOMMEN / WIEDER / SCHNEE / SCHNEIEN / SONNTAG / KOMMEN (ix / evening / come / again / snow / snowing / sunday / come) |
| | LLM Gloss ($\tilde{\mathcal{G}}_{\mathcal{V}}$) | ABEND / WEST / SCHNEE / KOMMEN / SONNTAG / KOMMEN (evening / west / snow / come / sunday / come) |
| | Frame-wise Gloss | ABEND / ABEND / WEST / KOMMEN / KOMMEN / WEST / SCHNEE / SCHNEE / SCHNEE / SONNTAG / DEUTSCHLAND / REGION / KOMMEN / KOMMEN / REGION / REGION |
| | Ref Gloss ($\tilde{\mathcal{G}}_{\mathcal{V},ref}$) | ABEND / WEST / KOMMEN / WEST / SCHNEE / SONNTAG / REGION / KOMMEN / REGION |
| | **Reordered Gloss** ($\tilde{\mathcal{G}}_{\mathcal{V},target}$) | ABEND / WEST / KOMMEN / SCHNEE / SONNTAG / KOMMEN (evening / west / come / snow / sunday / come) |
| (2) | **Text** ($\mathcal{S}$) | feuchte kalte luft strömt zu uns nach deutschland. (Moist cold air flows to us into germany.) |
| | **True Gloss** ($\mathcal{G}$) | TROCKEN / KALT / LUFT / KOMMEN / DEUTSCHLAND (dry / cold / air / come / germany) |
| | LLM Gloss ($\tilde{\mathcal{G}}_{\mathcal{V}}$) | KALT / LUFT / DEUTSCHLAND / KOMMEN (cold / air / germany / come) |
| | Frame-wise Gloss | KALT / KALT / KALT / LUFT / LUFT / KOMMEN / KOMMEN / KOMMEN / DEUTSCHLAND / DEUTSCHLAND |
| | Ref Gloss ($\tilde{\mathcal{G}}_{\mathcal{V},ref}$) | KALT / LUFT / KOMMEN / DEUTSCHLAND |
| | **Reordered Gloss** ($\tilde{\mathcal{G}}_{\mathcal{V},target}$) | KALT / LUFT / KOMMEN / DEUTSCHLAND (cold / air / come / germany) |
| (3) | **Text** ($\mathcal{S}$) | der donnerstag beginnt oft freundlich später zieht von westen regen heran. (Thursday begins often friendly later rain moves in from the west.) |
| | **True Gloss** ($\mathcal{G}$) | DONNERSTAG / FREUNDLICH / SONNE / DANN / SPAETER / KOMMEN / REGEN (thursday / friendly / sun / then / later / come / rain) |
| | LLM Gloss ($\tilde{\mathcal{G}}_{\mathcal{V}}$) | DONNERSTAG / FREUNDLICH / SPÄTER / WEST / REGEN / KOMMEN (thursday / friendly / later / west / rain / come) |
| | Frame-wise Gloss | DONNERSTAG / FREUNDLICH / SPÄTER / SPÄTER / WEST / KOMMEN / REGEN / REGEN / REGEN / NUR |
| | Ref Gloss ($\tilde{\mathcal{G}}_{\mathcal{V},ref}$) | DONNERSTAG / FREUNDLICH / SPÄTER / WEST / KOMMEN / REGEN / NUR |
| | **Reordered Gloss** ($\tilde{\mathcal{G}}_{\mathcal{V},target}$) | DONNERSTAG / FREUNDLICH / SPÄTER / WEST / KOMMEN / REGEN (thursday / friendly / later / west / come / rain) |
| (4) | **Text** ($\mathcal{S}$) | und nun die wettervorhersage für morgen samstag den fünfzehnten august. (And now the weather forecast for tomorrow, Saturday the fifteenth of August.) |
| | **True Gloss** ($\mathcal{G}$) | JETZT / WIE-AUSSEHEN / WETTER / MORGEN / SAMSTAG / FUENFZEHN / AUGUST (now / how-look / weather / tomorrow / saturday / fifteen / august) |
| | LLM Gloss ($\tilde{\mathcal{G}}_{\mathcal{V}}$) | MORGEN / WETTER / SAMSTAG / FÜNFZEHN / AUGUST (tomorrow / weather / saturday / fifteenth / august) |
| | Frame-wise Gloss | WETTER / WETTER / MORGEN / MORGEN / MORGEN / MORGEN / SAMSTAG / WETTER / WETTER / WETTER / WETTER / AUGUST |
| | Ref Gloss ($\tilde{\mathcal{G}}_{\mathcal{V},ref}$) | WETTER / MORGEN / SAMSTAG / WETTER / AUGUST |
| | **Reordered Gloss** ($\tilde{\mathcal{G}}_{\mathcal{V},target}$) | WETTER / MORGEN / SAMSTAG / FÜNFZEHN / AUGUST (weather / tomorrow / saturday / fifteenth / august) |
| (5) | **Text** ($\mathcal{S}$) | der heutige tag zwischen hitze im osten und abkühlung im westen. (Today is a day between heat in the east and cooling in the west.) |
| | **True Gloss** ($\mathcal{G}$) | HEISS / HIER / OST / REGION / KUEHL (hot / here / east / region / cool) |
| | LLM Gloss ($\tilde{\mathcal{G}}_{\mathcal{V}}$) | HEUTE / OST / HEISS / WEST / KALT (today / east / hot / west / cold) |
| | Frame-wise Gloss | AUCH / HEISS / HEISS / OST / OST / HEUTE / HEUTE / WEST / WEST / KALT / KALT / KOMMEN |
| | Ref Gloss ($\tilde{\mathcal{G}}_{\mathcal{V},ref}$) | HEISS / OST / HEUTE / WEST / KALT |
| | **Reordered Gloss** ($\tilde{\mathcal{G}}_{\mathcal{V},target}$) | HEISS / OST / HEUTE / WEST / KALT (hot / east / today / west / cold) |
| (6) | **Text** ($\mathcal{S}$) | am montag teils sonne teils wolken und einzelne gewitterschauer. (On Monday partly sunny partly cloudy and isolated thunderstorms.) |
| | **True Gloss** ($\mathcal{G}$) | MONTAG / SONNE / WOLKE / WECHSELHAFT / KOENNEN / GEWITTER (monday / sun / cloud / changeable / can / thunderstorm) |
| | LLM Gloss ($\tilde{\mathcal{G}}_{\mathcal{V}}$) | MONTAG / TEILS / SONNE / TEILS / WOLKE / UND / GEWITTER / KOENNEN (monday / partly / sun / partly / cloud / and / thunderstorm / can) |
| | Frame-wise Gloss | MONTAG / MONTAG / SONNE / UND / WOLKE / WOLKE / WOLKE / AUCH / KOENNEN / KOENNEN / GEWITTER / GEWITTER / GEWITTER |
| | Ref Gloss ($\tilde{\mathcal{G}}_{\mathcal{V},ref}$) | MONTAG / SONNE / UND / WOLKE / KOENNEN / GEWITTER |
| | **Reordered Gloss** ($\tilde{\mathcal{G}}_{\mathcal{V},target}$) | MONTAG / TEILS / SONNE / TEILS / UND / WOLKE / KOENNEN / GEWITTER (monday / partly / sun / partly / and / cloud / can / thunderstorm) |
| (7) | **Text** ($\mathcal{S}$) | örtlich muss mit bodenfrost gerechnet werden. (Locally ground frost must be expected.) |
| | **True Gloss** ($\mathcal{G}$) | ORT / MOEGLICH / BODEN / FROST (place / possible / ground / frost) |
| | LLM Gloss ($\tilde{\mathcal{G}}_{\mathcal{V}}$) | ORT / FROST / BODEN (place / frost / ground) |
| | Frame-wise Gloss | ORT / KOENNEN / BODEN / FROST / KOENNEN |
| | Ref Gloss ($\tilde{\mathcal{G}}_{\mathcal{V},ref}$) | ORT / BODEN / FROST |
| | **Reordered Gloss** ($\tilde{\mathcal{G}}_{\mathcal{V},target}$) | ORT / BODEN / FROST (place / ground / frost) |

Table 14: More examples of LLM-generated pseudo gloss and corresponding reordering operation on Phoenix14T.

| (1) | **Reference** | And that's a great vital point technique for women's self defense. |
| | Tarrés *et al.* [48] | It's really a great point for women's self defense. |
| | Uthus *et al.* [51] | It's really great for women's self defense. |
| | SSVP-SLT [45] | This is a really great point for women's self defense. |
| | PGG-SLT (ours) | These are great things for women's self defense. |
| (2) | **Reference** | In this clip I'm going to show you how to tape your cables down. |
| | Tarrés *et al.* [48] | In this clip I'm going to show you how to improve push ups. |
| | Uthus *et al.* [51] | In this clip we're going to show you how to cut a piece of clay. |
| | SSVP-SLT [45] | In this clip I'm going to show you how to clip the cable, the cable. |
| | PGG-SLT (ours) | In this clip I'm going to show you how to name the clipper wire. |
| (3) | **Reference** | In this segment we're going to talk about how to load your still for distillation of lavender essential oil. |
| | Tarrés *et al.* [48] | Ok, in this clip, we're going to talk about how to fold the ink for the lid of the oil. |
| | Uthus *et al.* [51] | In this clip we're going to talk about how to feed a set of baiting lizards for a lava field oil. |
| | SSVP-SLT [45] | In this clip we're going to talk about how to feed the trail for draining clean for laborer oil. |
| | PGG-SLT (ours) | In this clip we're going to talk about how to smooth out the sidewall for the dust wash for the lavatory oil. |
| (4) | **Reference** | You are dancing, and now you are going to need the veil and you are going to just grab the veil as far as possible. |
| | Tarrés *et al.* [48] | So, once you're belly dancing, once you've got to have the strap, you're going to need to grab the thumb, and try to avoid it. |
| | Uthus *et al.* [51] | Their hopping and dancing is now, they're going to need their squat and squat and they're going to be able to move independently. |
| | SSVP-SLT [45] | So that she's going to get her hips up as far as she can, and now she's going to lift her head up as far as possible. |
| | PGG-SLT (ours) | Your belly dancing right now should be going up and down as far as you can. |
| (5) | **Reference** | But if you have to setup a new campfire, there's two ways to do it in a very low impact; one is with a mound fire, which we should in the campfire segment earlier and the other way to setup a low impact campfire is to have a fire pan, which is just a steel pan like the top of a trash can. |
| | Tarrés *et al.* [48] | And other thing I'm going to talk to you is a little bit more space, a space that's what it's going to do, it's kind of a quick, and then I don't want to take a spray skirt off, and then I don't want it to take it to the top of it. |
| | Uthus *et al.* [51] | But if you have to set up a new campfire, there are two ways to do a low impact fire, one is a cone fire, which we have to do in the tent earlier, and the other one is to set up a campfire in a fire pan. |
| | SSVP-SLT [45] | But if you have to set up a new campfire, this is one way to do it in a low impact. One is a monk fire. One is a campfire. The other one is to set a campfire in a campfire. That's just a post like the top of the post. |
| | PGG-SLT (ours) | But if we're going to set up a new campfire, there's two ways to do a low impact fire which is a marking fire which we should do in the center of the campfire, and another one is just to set a fire pan, and that's just the top of the pan. |
| (6) | **Reference** | So, this is a very important part of the process. |
| | Tarrés *et al.* [48] | It's a very important part of the process. |
| | Uthus *et al.* [51] | Alright, let's get started. |
| | SSVP-SLT [45] | It's an important part of the process. |
| | PGG-SLT (ours) | This is a good part of the process. |

Table 15: Qualitative translation examples from our mBART based version compared to Tarrés *et al.* [48], Uthus *et al.* [51], SSVP-SLT [45], and the reference translations on How2Sign [16]. The examples were picked from the How2Sign test set by [48]. Although the BLEU4 score of our method is slightly lower than that of SSVP-SLT, our model's predictions remain largely on-topic, similar to SSVP-SLT. Moreover, all current models can struggle with repetitions and sign mix-ups in many cases.

| (1) | **Reference** | Beautiful. |
| | Prediction | Beautiful. |
| (2) | **Reference** | One more time. |
| | Prediction | One more time. |
| (3) | **Reference** | Inhaling all the way up. |
| | Prediction | And he breathes all the way up. |
| (4) | **Reference** | Drag that energy down to the heart. |
| | Prediction | Draw the energy down to the heart. |
| (5) | **Reference** | So I'm shuffling this deck at the start of this segment because instead of laying out bad hands, or Jacks to Open, I'm going to show them to you right off the deck. |
| | Prediction | And I'm going to talk about the amount that I'm going to be spending to get started because I'm going to need to get rid of the seeds, I'm going to get rid of the bad soil in the open, and I'm going to show you how to draw it. |
| (6) | **Reference** | And so let me show you how to set it up. |
| | Prediction | Maybe I should show you how to set this up. |
| (7) | **Reference** | I am going to take these parts of my hand right there and I am going to squeeze the clay and push it up. |
| | Prediction | We're going to take our bowl and remove some of the clay and we're going to squeeze our bowl up. |
| (8) | **Reference** | Make sure that when you are placing your veil, you need to grab your veil like this. |
| | Prediction | So to make sure that when you set up your tent you should take a hat like this. |
| (9) | **Reference** | So again, just practice plunking out notes, singing some scales, and some different notes, and sooner or later you will have a melody. |
| | Prediction | It just takes a little practice and you can get some slims in different parts of the body, and eventually they can become very slippery. |
| (10) | **Reference** | We'll have Emily standing about hip width apart. |
| | Prediction | So daniel's going to stand with his feet about hip width apart. |
| (11) | **Reference** | Now I'm ready for my legs. |
| | Prediction | Now we're ready for the legs. |
| (12) | **Reference** | I'll start with my upper body even though my legs are going to be the ones that will really be the focus. |
| | Prediction | We're going to start with our higher body kind of whatever the leg is that's really great. |
| (13) | **Reference** | If you have a queen, and let's say you have a ten, it's irrelevant what the third card is 'cause it's already better than a queen-six-four. |
| | Prediction | If you're a queen of 10 let's say you're a queen of 3 it's much better than a queen of 4. |
| (14) | **Reference** | Typically if you have higher than a queen-six-four, you will want to stay so you'll want to put your bet in the play column. |
| | Prediction | If it's a little bit longer than your sleeve, you might want to stay with it and that's how you play line up for your sleeve. |
| (15) | **Reference** | If it's under that, you want to fold, you will lose your ante. |
| | Prediction | If it's below your comfort level you can wear it while you're dancing. |
| (16) | **Reference** | If you have a queen, but your two cards are less then the dealers, you will lose and the dealer will take both of your bets. |
| | Prediction | If you're a queen, you can only have two cards, so you have to either lose your opponent's hand or you can have the rhythm of losing your opponent's hand. |
| (17) | **Reference** | Again I want to make faces. |
| | Prediction | And again you just want to make sure you're putting it right on the face. |

Table 16: More qualitative translation examples from our mBART based version on How2Sign [16].

