# OpenReview forum: "Bridging Sign and Spoken Languages: Pseudo Gloss Generation for Sign Language Translation"
_NeurIPS.cc/2025/Conference — NeurIPS 2025 poster_

### Official Review · Reviewer_tD2m · 2025-06-26

**Clarity:** 3
**Significance:** 3
**Originality:** 3
**Rating:** 4
**Confidence:** 4

**Summary:**

This paper introduces a novel gloss-free pseudo gloss generation framework for Sign Language Translation (SLT) that addresses the limitations of relying on costly and scarce human-annotated gloss labels. The proposed method, PGG-SLT, leverages Large Language Models (LLMs) to produce "pseudo glosses" from spoken language text using in-context learning with a few example pairs. To overcome the issue of LLM-generated glosses not aligning temporally with sign videos, a weakly supervised learning process is introduced to reorder these pseudo glosses.

**Questions:**

See weaknesses.

**Ethical Concerns:**

["NO or VERY MINOR ethics concerns only"]

**Final Justification:**

I am satisfied with the paper. And since the work has not reached SOTA and weakness 2 in my review is not solved, I will keep my score.

**Limitations:**

Since, for various reasons(Like missing large-scale pertaining), the method hasn't reached SOTA performance, I recommend adding a limitation section and listing some potential directions for future work.

**Paper Formatting Concerns:**

No concerns.

**Quality:**

3

**Strengths And Weaknesses:**

### Strengths:
1. The paper is clearly written and easy to follow.
2. The proposed method for pseudo-labeling and weakly supervised order refinement.
3. The experiment is solid and promising. The ablation analysis is thorough, examining various important design choices in depth.
### Weaknesses:
1. The main advantage of the gloss-free method is the potential for scaling up data.  The paper would be more complete if an experiment about the scalability were conducted, for example, by improving performance through introducing extra data or showing the capability to generalize over different domains.
2. I am a little doubtful whether the order is a real problem. From table(d), the order refinement results in about a 0.5 to 1 BLEU4 difference. I am curious whether this difference can be narrowed if we use a structure that does not require strict alignment, such as RNNT and the Transformer decoder, for predicting the gloss.

---

> ### Author Rebuttal · Authors · 2025-07-31
>
> We sincerely thank the reviewer for providing supportive comments and acknowledging the strengths of our work. We respond to your questions as follows:
>
> >W1. About scaling up data.
>
> Thanks for your comments. We agree that evaluating on even larger web-scale datasets (e.g., YouTubeSL25) is an important next step. We will explicitly highlight this as a key direction for future work in the Limitations and Future Work section. To date, our experimental validation on scaled-up datasets has been limited to Phoenix14T and How2Sign. That said, the How2Sign dataset (being approximately five times the size of Phoenix14T) does provide preliminary evidence of our method’s scalability within a certain range. As for the generalization over different domains, we further conduct the experiments on CSL-Daily dataset, which is based on Chinese, expanding beyond the German and English contexts. The results are shown below:
>
> |Model||Dev Set|||Test Set||
> |:-:|:-:|:-:|:-:|:-:|:-:|:-:|
> ||BLEU1|BLEU4|ROUGE|BLEU1|BLEU4|ROUGE|
> |TS-SLT|58.24|29.18|57.81|58.64|29.55|58.62|
> |GFSLT-VLP*|39.20|11.07|36.70|39.37|11.00|36.44|
> |Sign2GPT*|-|-|-|41.75|15.40|42.36|
> |SignLLM*|42.45|12.23|39.18|39.55|15.75|39.91|
> |VAP*|53.31|23.84|51.19|52.98|23.65|51.09|
> |Ours*|53.58|24.05|52.34|53.29|23.70|52.88|
>
> >W2. RNNT and the Transformer decoder for predicting the gloss.
>
> This is indeed an intriguing direction. In our current setting, we don’t modify the baseline SLT model itself. This makes the generate-then-reorder a practical and effective solution within the existing experimental setup, as it can be seamlessly integrated without requiring architectural adjustments to the underlying translation framework. We will also discuss this in our Future Work section.
>
> >Q1 Adding a limitation section.
>
> Thanks for your valuable suggestion. We will add following discussion to an additional section Limitation and Future Work:
>
> One limitation of this work is that we only scaled the dataset from Phoenix14T to How2Sign, and the performance of our proposed PGG-SLT when incorporating additional training data remains unvalidated. As shown in Table 5, two other methods that leverage large-scale datasets for pre-training achieve slightly better results than our model. Moving forward, we plan to evaluate the effectiveness of our pseudo-gloss generation method on larger ASL benchmarks. Additionally, we will explore whether mixing diverse large-scale datasets (e.g., ASL and BSL) can further benefit SLT model training, given their shared connection to English as a spoken language.

---

### Official Review · Reviewer_UNMc · 2025-07-03

**Clarity:** 3
**Significance:** 3
**Originality:** 3
**Rating:** 5
**Confidence:** 3

**Summary:**

This work explores generation of pseudo-glosses to support sign language translation in settings without gloss annotation, using LLMs for pseudo gloss generation

**Questions:**

Questions:
- The description of stage 2: gloss2sign is slightly confusing, unclear why we are going back to sign – could you clarify?
- German and English have different performance for mBART and Gemma, how do those impact corresponding glossing?
- Which sentences / example pairs are used to train the LLMs? How does this impact the performance?

Suggestions:
- Improved discussion of glossing (as suggested above)

**Ethical Concerns:**

["NO or VERY MINOR ethics concerns only"]

**Final Justification:**

I recommend Accept. The proposed changes (additions to the limitations and future work section, a more nuanced discussion of glossing, and addition to the appendix) satisfies my concerns.

**Limitations:**

See above related to glossing.

**Quality:**

3

**Strengths And Weaknesses:**

As data annotation (glossing) for sign languages is fairly laborious task, the gloss-free sign language translation task explored here is very valuable, particularly for lower resource settings. Notably their results show performance on par with gloss-based approaches. Their pseudo gloss-reordering approach is interesting, the authors conduct a comprehensive set of experiments and the work is well written.

My main concerns for this work are related to some of the nuances with claims made and framing, particularly related to how glosses are defined and described.  In the introduction, glossing is framed as something specific to sign languages – but in fact, glosses are an intermediary representation used in linguistics when considering translations between languages. Therefore, glosses are not a written form of sign languages (there do exist sign language writing systems, not adopted widely).
Additionally, many of the issues described about the different syntax and ordering between signed and spoken languages are pretty much translation aspects applicable to translation between all languages (e.g., between two spoken languages).  I think accurately framing the problem is necessary to evaluate claims but also to situate the work correctly amongst other related research areas.

Glossing for sign language is typically done with reference to sign language, so as to correctly mark the signs present in the sentence. In this work, they instead use the  english translations to approximate the glosses present in the signed sentence. My main concern with this approach is that it seems that the expressiveness of sign language is greatly reduced in the representations. For example, the same English sentence may be signed many different ways in ASL – using different signs, perhaps incorporating classifiers, 3D space, idioms and other unique linguistic features. Using the english translation for glossing then would fail to capture any of this linguistic diversity (e.g., consider many ASL idioms like TOUCH-FINISH, TRAIN-GONE, that are not represented in corresponding english translation at all). Should these learned models and representations ever be used to in sign language generation, the produced sign language would be very homogenized, and bring in spoken language bias.

I am also additionally skeptical of the claim that LLMs are surrogate experts proficient in sign languages – what is the basis of this claim? While LLMs might be good at tasks of “filtering out unimportant words from text” that is not glossing which linguists do on while referring to the sign language video alone. The results are further used to claim that the generated pseudo glosses approximate real gloss annotations – but the experiments do not substantiate this claim. It would require a comparison of generated pseudo glosses to existing glosses. Perhaps a more accurate claim would be that pseudo glosses have comparable value in supporting sign language translation. I believe that the distinction is important, given that glosses have use across different languages SLRGT and linguistic tasks.

Overall, I do think that this work is valuable – I would love to see a more accurate discussion of glossing, and limitations of this gloss-free / pseudo-glossing approach (e.g., linguistic limitations, spoken language bias etc).

---

> ### Author Rebuttal · Authors · 2025-07-31
>
> We sincerely thank the reviewer for providing supportive comments and acknowledging the strengths of our work. We respond to your questions as follows:
>
> > Discussion about the glossing.
>
> Thank you for pointing out this point. We agree that it addresses a fundamental aspect of the sign language domain.
>
> First, regarding glossing: Indeed, reducing certain action video frames to a few "words" inherently limits the expressiveness of sign language in such representations. This is why resources like 2M-Flores-ASL incorporate markers such as "+++" to indicate repeated actions, "cl:" to denote specific handshape classifiers, as well as markers like / /WHAT\ \ and / /WHY\ \ to represent NMMs (with / / and \ \ indicating their span) in their dataset. Further, sign language exhibits substantial variability, different sign sequences can convey the same meaning, much like how spoken English uses diverse vocabulary for identical concepts. Crucially, text (spoken language) alone cannot perfectly reverse-engineer the specific visual details (e.g., precise hand movements, spatial configurations) captured in video. That said, given current constraints (e.g., dataset scale and model capabilities), we think that our current relatively "simple" glossing approach offers practical advantages: it lowers the learning barrier for models and facilitates more measurable improvements in SLT performance (e.g., BLEU scores). We acknowledge the validity of your concerns, but implementing a more nuanced, expressive representation remains challenging with existing model and data. To address this, we will add an additional section in Limitations and Future Work to discuss these trade-offs, along with avenues for advancing glossing methods to better capture sign language’s rich expressivity.
>
> Regarding "filtering out unimportant words from text," we agree that a true gloss translates only from the sign language video. In this paper, our key point is that core video content in SLT is likely reflected in the corresponding spoken text, playing to LLMs' strength in extracting key info. Table 6a shows our generated glosses outperform Sign2GPT’s (POS row) via WER and BlEU1/BLEU4 comparisons with true glosses. In our next version, we’ll clarify our outputs are not video-derived "true" glosses, but argue these pseudo-glosses have comparable value for SLT, with discussion on glossing nuances in the related work section.
>
> > Q1. The description of stage 2: gloss2sign.
>
> Sorry for the confusion. Figure 1 should be "gloss2text," consistent with Line 152.
>
> > Q2. German and English have different performance for mBART and Gemma, how do those impact corresponding glossing?
>
> For current glossing (specifically, generating LLM-based pseudo-glosses), we employ the Gemini 1.5 Pro model which is trained in a multilingual setting, it natively understands German and English inputs, including the example text-gloss pairs. For the mBART- and Gemma-based SLT models, their respective tokenizers are used to process German and English, respectively.
>
> > Q3. Which sentences / example pairs are used to train the LLMs? How does this impact the performance?
>
> This is a good question. We randomly select example pairs to prompt the LLM for pseudo-gloss generation, and we didn’t fine-tune the LLM, relying solely on these examples as prompts. To date, we have not explored how different pair selections might affect results, but we have conducted experiments comparing text-gloss vs. gloss-text pair formats, e.g., “Example1: text: [] gloss: []” vs. “Example 1: gloss: [] text: []”. For gloss quality, using text-gloss pairs as prompts yields pseudo-glosses closer to true glosses (with better BLEU1 and BLEU4 scores) than gloss-text prompts. We will add relevant discussion in the paper’s appendix.

---

> > ### Comment · Reviewer_UNMc · 2025-08-05
> >
> > Thank you for your thoughtful rebuttal and engagement with the review! The proposed changes (additions to the limitations and future work section, a more nuanced discussion of glossing, and addition to the appendix) satisfies my concerns. A small follow-up regarding Q2: While Gemini might be trained in multilingual settings, do you know if the performance for German vs. English is equally good / comparable?

---

> ### Author Response · Authors · 2025-08-06
>
> Thanks for your reply. Your follow-up question is quite insightful.
>
> Regarding quantifiable data, Gemini 1.5 Pro presents results for WMT23 (sentence-level machine translation) and MGSM (multilingual math reasoning) in Sec 6.1.4 of their technical report "Gemini 1.5: Unlocking multimodal understanding across millions of tokens of context." Specifically, the WMT23 task includes a series of document-level evaluations for German-English and English-German translation, where Gemini 1.5 Pro achieved an overall 1-shot performance exceeding 75.
>
> Additional quantifiable evidence can be found on the Language Model Text Arena leaderboard (https://huggingface.co/spaces/lmarena-ai/lmarena-leaderboard), which provides comprehensive evaluations across multiple languages. As of now, Gemini 1.5 Pro scores 1313 in the German category (ranking 30th) and 1354 in the English category (ranking 43rd). These consistent standings across both languages suggest that Gemini maintains a well-balanced multilingual performance. It’s also worth noting that the latest version of the Gemini model (Gemini 2.5 Pro) currently ranks 1st on the same leaderboard. While this latest version is not used in our current paper, it may reflect the broader trajectory and capability of the Gemini family in multilingual understanding.

---

### Official Review · Reviewer_ujwE · 2025-07-03

**Clarity:** 3
**Significance:** 3
**Originality:** 3
**Rating:** 4
**Confidence:** 5

**Summary:**

This paper aims to alleviate the reliance on costly gloss annotation with the proposed pseudo gloss generation framework. It starts by prompting LLM with a few samples and utilizes weakly-supervised ordering for better alignment. Experiments are conducted on two datasets to validate the effectiveness.

**Questions:**

The questions are raised in weakness part.

**Ethical Concerns:**

["NO or VERY MINOR ethics concerns only"]

**Final Justification:**

Thank the authors for their detailed rebuttal. It addressed most of my concerns. I confirm to keep my initial score.

**Limitations:**

Please discuss the limitations in a section.

**Quality:**

3

**Strengths And Weaknesses:**

Strengths:

1.	The idea is clearly presented.
2.	The paper is well-organized.
3.	The experiment results look promising.

Weaknesses:

1.	The authors should ensure the rigor of language expression. It should not be claimed as a gloss-free generation framework since it leverages several gloss-text pairs as the prompt of LLMs.
2.	It is suggested to discuss/demonstrate whether the proposed method could be generalizable to more sign languages like CSL.
3.	How do authors choose the gloss-text pairs to ensure more diversity and less bias?
4.	From Table 6a, it seems that the upper bound of gloss generation is still low but does not have a great impact on the final SLT performance. Could the authors provide more explanation on it?
5.	Could the authors analyze the failure cases?
6.	Some related SLT pre-training works are suggested to cite, SignBERT+, SHuBERT…

---

> ### Author Rebuttal · Authors · 2025-07-31
>
> We sincerely thank the reviewer for providing supportive comments and acknowledging the strengths of our work. We respond to your questions as follows:
>
> >Q1. The authors should ensure the rigor of language expression. It should not be claimed as a gloss-free generation framework since it leverages several gloss-text pairs as the prompt of LLMs.
>
> Thank you for your advice. You raise an excellent point regarding the precision of terminology. We agree that the term "gloss-free" can be misleading, as our method does leverage a small number of example gloss-text pairs. We will clarify this when introducing our method (e.g., Line 62), and revise relevant sections of the paper to better contextualize our approach. Specifically, for the PHOENIX14T dataset, our approach is not gloss-free, as we use available gloss annotations during training. However, for datasets such as How2Sign, where no gloss annotations are inherently provided, we instead utilize gloss-text examples from other datasets. In this case, we consider our method to be gloss-free with respect to the target dataset. Our original intention was to emphasize that our approach eliminates the need for gloss annotations across the vast majority of the training data, particularly for domains where such annotations are unavailable or costly to obtain. To more accurately reflect this, we will revise the paper to replace "gloss-free" with more precise terms such as "annotation-efficient" or "minimally supervised", as appropriate. Thank you for this important clarification—it will help us better articulate the strengths and limitations of our methodology.
>
> >Q2. It is suggested to discuss/demonstrate whether the proposed method could be generalizable to more sign languages like CSL.
>
> We further evaluate our method (mBART-based version) on CSL-Daily dataset, * means the Gloss-free method.
>
> |Model||Dev Set|||Test Set||
> |:-:|:-:|:-:|:-:|:-:|:-:|:-:|
> ||BLEU1|BLEU4|ROUGE|BLEU1|BLEU4|ROUGE|
> |TS-SLT|58.24|29.18|57.81|58.64|29.55|58.62|
> |GFSLT-VLP*|39.20|11.07|36.70|39.37|11.00|36.44|
> |Sign2GPT*|-|-|-|41.75|15.40|42.36|
> |SignLLM*|42.45|12.23|39.18|39.55|15.75|39.91|
> |VAP*|53.31|23.84|51.19|52.98|23.65|51.09|
> |Ours*|53.58|24.05|52.34|53.29|23.70|52.88|
>
> >Q3. How do authors choose the gloss-text pairs to ensure more diversity and less bias?
>
> This is a good question, and one we haven’t yet considered. Currently, we use random selection, which we will explicitly note and discuss in our Future Work section.
> To date, we have only conducted experiments comparing text-gloss vs. gloss-text pair formats in prompt, e.g., “Example1: text: [] gloss: []” vs. “Example 1: gloss: [] text: []”. For generated gloss quality, using text-gloss pairs as prompts yields pseudo-glosses closer to true glosses (with better BLEU1 and BLEU4 scores) than gloss-text prompts. We agree that prompt engineering is a critical factor in optimizing performance, and we will add a relevant discussion of this point in the paper’s appendix.
>
> >Q4. From Table 6a, it seems that the upper bound of gloss generation is still low but does not have a great impact on the final SLT performance. Could the authors provide more explanation on it?
>
> Thank you for your valuable comments. The key insight behind our approach is that even imperfect pseudo-glosses provide a far more informative training signal than having no glosses at all, as they help bridge the modality gap between sign language and spoken language.
> Regarding concerns about the quality ceiling of gloss generation, the 700 pairs of data (last row in Table 6a) represent less than 10% of the entire training set. We believe gloss generation quality will continue to improve with a larger proportion of samples used. However, we have also observed that even when test samples were referenced during the in-context phase, the LLM still failed to generate 100% accurate glosses in some cases. We attribute this to two factors: current LLMs are not specifically trained for such tasks, and glosses inherently have a “misaligned” relationship with spoken language. Additionally, the logical reasoning capabilities of current LLMs are not flawless (e.g., RAG systems cannot achieve 100% accuracy), resulting in certain generation limitations.
>
> As for the impact on final SLT performance, Table 6b still shows some improvements. Since this paper does not modify the training method of the baseline model, the upper bound of the gloss-free approach remains consistent with previous gloss-based methods.
>
> >Q5. Could the authors analyze the failure cases?
>
> We present here several typical error cases in How2Sign dataset here:
>
> - Reference：We’ll have Emily standing about hip width apart.
>
> - Prediction：So daniel’s going to stand with his feet about hip width apart.
>
> - Reference：Now I’m ready for my legs.
>
> - Prediction：Now we’re ready for the legs.
>
> The example pairs we use convert demonstrative pronouns to "ix" and render names in finger spelling form (e.g., "Emily" becomes "E-M-I-L-Y"). However, in practice, some videos may not spell out names via finger spelling. This mismatch leads to obvious referential errors in the model’s outputs.
>
> - Reference: So again, just practice plunking out notes, singing some scales, and some different notes, and sooner or later you will have a melody.
>
> - Prediction: It just takes a little practice and you can get some slims in different parts of the
> body, and eventually they can become very slippery.
>
> In this case, the model appears to have missed the core meaning of the input. As a result, it resolved the uncertain or ambiguous signs into semantically unrelated or nonsensical outputs. In How2Sign, there are also more complex scenarios, typically involving lengthy videos and texts—many of which result in translation errors. This suggests that current models may have limitations when handling longer text inputs and extended video frames.
>
> >Q6. Some related SLT pre-training works are suggested to cite, SignBERT+, SHuBERT…
>
> Thank you for pointing this out. We will incorporate discussions of these papers into Section 2 (Related Work):
>
> In addition, another research direction focuses on enhancing SLT and fingerspelling model performance through pre-training. SignBERT first proposed a self-supervised pre-training followed by downstream-task fine-tuning framework. Building on this, SignBERT+ introduced a self-supervised pre-training approach that models hand poses as visual tokens using multi-level masked modeling strategies and incorporates model-aware hand priors to capture hierarchical context. Meanwhile, SHuBERT adapted masked token prediction objectives to multi-stream visual sign language inputs, learning to predict multiple targets corresponding to clustered hand, face, and body pose streams.

---

> > ### Comment · Reviewer_ujwE · 2025-08-06
> >
> > Thank the authors for their detailed rebuttal. It addressed most of my concerns. Please incorporate these response during revision. I will keep my initial score.

---

> > > ### Author Response · Authors · 2025-08-07
> > >
> > > We are glad to hear that we were able to address your concerns, We will incorporate the response to further improve our final version.

---

> ### Author Response · Authors · 2025-08-06
> **Looking forward to your feedback in discussion period**
>
> Dear Reviewer ujwE,
>
> As the discussion deadline is approaching, we would like to kindly follow up on our recent rebuttal. We greatly appreciate the valuable feedback provided so far, and we hope that our revisions and responses have addressed your concerns effectively. If there are any remaining questions or points that require further clarification, we would be more than happy to discuss with you. Your guidance would be immensely helpful as we move toward the final decision.
>
> Thank you again for your time and consideration.

---

### Official Review · Reviewer_exiF · 2025-07-03

**Clarity:** 2
**Significance:** 3
**Originality:** 3
**Rating:** 4
**Confidence:** 4

**Summary:**

This paper addresses the challenge of Sign Language Translation (SLT), which converts sign language videos into spoken language text. Traditional SLT methods rely on costly, expert-annotated glosses as intermediate supervision, limiting scalability. To handle this, PGG-SLT is proposed in this paper, a gloss-free framework that leverages Large Language Models (LLMs) to generate pseudo glosses from spoken language text. These pseudo glosses are refined via a weakly supervised reordering mechanism to better align with sign video sequences, including three training stages, video-to-pseudo-gloss recognition, pseudo-gloss-to-text translation, and end-to-end fine-tuning. PGG-SLT provides a scalable, annotation-efficient solution for SLT by harnessing LLMs for pseudo gloss generation and weakly supervised alignment, achieving competitive accuracy without gloss supervision. The framework demonstrates strong generalization across languages and datasets.

**Questions:**

1.	How do LLM-generated pseudo-glosses capture sign language-specific linguistic features, such as spatial verbs, non-manual markers, that diverge fundamentally from spoken language syntax? Quantitative analysis comparing pseudo-glosses with ground truth for linguistic features, including missing classifiers, incorrect verbs, is necessary.
2.	Does the method maintain performance on sign languages with radically different grammar, such as Chinese and Japanese, or noisy real-world videos? More tests and evaluations could be implemented on datasets based on different languages, like CSL.
3.	A few illustrative figures are contained in this paper, and the existing figures fail to effectively convey the core idea and explain the algorithm.

**Ethical Concerns:**

["NO or VERY MINOR ethics concerns only"]

**Final Justification:**

The authors' response has solved most of my concerns. Based on the response and other reviewers' comments, I would raise my rating to Borderline accept.

**Limitations:**

yes

**Paper Formatting Concerns:**

Pseudo-code is directly inserted into the article, and figures are embedded within the text without separate paragraphs.

**Quality:**

3

**Strengths And Weaknesses:**

Strengths
+A Gloss-Free method is proposed to bypass the costly need for expert-annotated glosses, leveraging LLMs to generate pseudo glosses from spoken language text, which addresses a critical scalability bottleneck in SLT.
+A weakly supervised video-guided refinement process is introduced, using multi-label classification and temporal smoothing, to align LLM-generated pseudo glosses with sign sequences, enabling effective CTC loss supervision.
+The results outperform gloss-free methods on Phoenix14T, specifically, BLEU4 +5.0 over GFSLT-VLP, and +6.1 BLEU4 over SSVP-SLT, closing the gap with gloss-based methods.
Weaknesses
-LLM-generated glosses inherit spoken language syntax, failing to fully capture sign language-specific grammar. Reordering mitigates but does not eliminate misalignment.
-Performance highly depends on LLM capabilities, but smaller LLMs might underperform larger counterparts.
-Experiments are limited to German (Phoenix14T) and English (How2Sign) sign languages. Robustness regarding non-Western or low-resource sign languages is untested.

---

> ### Author Rebuttal · Authors · 2025-07-31
>
> We sincerely thank the reviewer for providing valuable comments. We respond to your questions and provide clarifications as follows:
>
> >Q1.1 How do LLM-generated pseudo-glosses capture sign language-specific linguistic features, such as spatial verbs, non-manual markers, that diverge fundamentally from spoken language syntax? & W1 LLM-generated glosses inherit spoken language syntax, failing to fully capture sign language-specific grammar.
>
> Thanks for your comments. We acknowledge that our current LLM is not yet capable of generating glosses that include non-manual markers or classifier annotations. For example, some glosses in the 2M-Flores-ASL dataset use "+++" to indicate repeated actions and "cl:" to denote specific handshape classifiers, as well as markers like / /WHAT\ \ and / /WHY\ \ to represent NMMs (with / / and \ \ indicating their span). We think this task is quite difficult without access to video information.
>
> In our study, most glossing requires only a single word to accurately describe what is being signed. The proposed LLM-generated glosses perform well in handling the unique grammatical structure of sign languages, which often differ significantly from spoken language word order, while effectively capturing some sign-specific patterns and filtering out spoken language artifacts. This leads to a more robust and scalable pipeline for automated sign language processing. We believe that, with sufficient text-gloss examples, LLMs can learn to produce such features through in-context learning, functioning as a dynamic "lookup table" that is continually refined by contextual cues. Our current focus is for the LLM to capture two key aspects: (1) differences in lexical grammar and (2) divergent expressive patterns. For example, the spoken phrase "In the north and west sometimes sun sometimes clouds" is glossed in sign language as "northwest / changeable / sun / cloud"—using "changeable" to condense the spoken meaning, while omitting spoken-specific prepositions. The LLM can filter out such spoken language elements, including prepositions and other structures tied to spoken language that are irrelevant to sign glosses. We show that a small set of examples explicitly containing these features allows LLM to infer unique sign language structures without explicit rule engineering. We observe performance improves with more examples, as the model refines its grasp of how sign language syntax diverges from spoken norms.
>
> >Q1.2 Quantitative analysis comparing pseudo-glosses with ground truth for linguistic features, including missing classifiers, incorrect verbs, is necessary.
>
> Thank you for your advice. Since the How2Sign dataset does not provide corresponding gloss annotations (including classifier markings), we currently offer only a lexical analysis, which serves as an essential first step toward the deeper linguistic analysis requested. We have conducted quantitative comparisons between the generated pseudo-glosses and the ground truth glosses, with the results summarized as follows:
>
> |Dataset| | |Phoenix14T| | | CSL-Daily |
> |:-:|:-:|:-:|:-:|:-:|:-:|:-:|
> |        | Overall | Noun | Numerical | Adjective | Verb | Overall |
> | Precision| 58.2 | 63.8 | 43.2       | 61.2    | 55.9    | 72.4  |
> | Recall     | 63.5 | 66.9 | 45.1       | 63.5    | 59.1    | 82.8  |
>
> For computational convenience, we first processed both the generated glosses and ground truth by removing duplicate words, then calculated the precision and recall between each generated gloss and its corresponding ground truth gloss (denoted as overall). Additionally, we used the spaCy natural language processing library to perform lemmatization and categorize words into several part-of-speech classes, and further analyzed the generation quality of several key categories.
>
> As shown in the table above,  our method achieves quite strong precision and recall, especially for key content words such as nouns. We will include the full analysis in the appendix. We acknowledge that this evaluation is conducted at the lexical level. A more in-depth analysis addressing missing classifiers or incorrect verb types, as you suggest, represents a valuable direction for future linguistic evaluation of sign language translation systems.
>
> > Q2. Does the method maintain performance on sign languages with radically different grammar, such as Chinese and Japanese, or noisy real-world videos? More tests and evaluations could be implemented on datasets based on different languages, like CSL.
>
> We further evaluate our method (mBART-based version) on CSL-Daily dataset, * means the Gloss-free method.
>
> |Model||Dev Set|||Test Set||
> |:-:|:-:|:-:|:-:|:-:|:-:|:-:|
> ||BLEU1|BLEU4|ROUGE|BLEU1|BLEU4|ROUGE|
> |TS-SLT|58.24|29.18|57.81|58.64|29.55|58.62|
> |GFSLT-VLP*|39.20|11.07|36.70|39.37|11.00|36.44|
> |Sign2GPT*|-|-|-|41.75|15.40|42.36|
> |SignLLM*|42.45|12.23|39.18|39.55|15.75|39.91|
> |VAP*|53.31|23.84|51.19|52.98|23.65|51.09|
> |Ours*|53.58|24.05|52.34|53.29|23.70|52.88|
>
>
> >Q3. A few illustrative figures are contained in this paper, and the existing figures fail to effectively convey the core idea and explain the algorithm.
>
> Thank you for this feedback. We agree that the figures can be improved. In our revision, we will create a new, comprehensive Figure 1 that visually integrates the entire pipeline: (1) Prompting the LLM with text-gloss pairs, (2) showing the initial LLM-generated gloss, (3) illustrating the video-guided refinement process from Algorithm 1, and (4) showing the final reordered gloss used for training. This will provide a much clearer, end-to-end overview of our core contribution.

---

> > ### Comment · Reviewer_exiF · 2025-08-07
> >
> > I appreciate the authors' response. It solves most of my concerns.  To better demonstrate the pipeline of the algorithm, I suggest that more comprehensive figures could be added to the paper to explain the authors’ core idea.

---

> > > ### Author Response · Authors · 2025-08-07
> > >
> > > We are glad to hear that we were able to address your concerns. In addition to the refined Figure 1, we will also include additional pseudo gloss generation figure into our final version.
> > >
> > > Thank you again for your time and suggestion.

---

> ### Author Response · Authors · 2025-08-06
> **Looking forward to your feedback in discussion period**
>
> Dear Reviewer exiF,
>
> As the discussion deadline is approaching, we would like to kindly follow up on our recent rebuttal. We greatly appreciate the valuable feedback provided so far, and we hope that our revisions and responses have addressed your concerns effectively. If there are any remaining questions or points that require further clarification, we would be more than happy to discuss with you. Your guidance would be immensely helpful as we move toward the final decision.
>
> Thank you again for your time and consideration.

---

### Decision · Program_Chairs · 2025-09-17

**Decision:**

Accept (poster)

**Comment:**

To reduce the costly-annotation in gloss labeling for sign language recognition, this submission proposes a gloss-free pseudo gloss generation method with a few example text-gloss pairs. The proposed method is very clear and easy to follow, and result is very promising. However, sign language as a low resource language, the result on sign language highly dependences the LLM. The submission received one borderline reject and three borderline accepts before rebuttal. However, after the deeply discussion between authors and reviewers, two reviewers increased their ratings, and this submission received one accept and three borderline accepts. All the reviewers reach a consensus of acceptance. The AC agree with the reviewers and recommend accepting this submission.